# Confounding-Robust Policy Evaluation in Infinite-Horizon Reinforcement Learning

**Nathan Kallus**
School of Operations Research and
Information Engineering
Cornell University and Cornell Tech
kallus@cornell.edu

**Angela Zhou**[*]
School of Operations Research and
Information Engineering
Cornell University and Cornell Tech
az434@cornell.edu

## Abstract

Off-policy evaluation of sequential decision policies from observational data is necessary in applications of batch reinforcement learning such as education and healthcare. In such settings, however, unobserved variables confound observed actions, rendering exact evaluation of new policies impossible, *i.e.*, unidentifiable. We develop a robust approach that estimates sharp bounds on the (unidentifiable) value of a given policy in an infinite-horizon problem given data from another policy with unobserved confounding, subject to a sensitivity model. We consider stationary unobserved confounding and compute bounds by optimizing over the set of all stationary state-occupancy ratios that agree with a new partially identified estimating equation and the sensitivity model. We prove convergence to the sharp bounds as we collect more confounded data. Although checking set membership is a linear program, the support function is given by a difficult nonconvex optimization problem. We develop approximations based on nonconvex projected gradient descent and demonstrate the resulting bounds empirically.

## 1 Introduction

Evaluation of sequential decision-making policies under uncertainty is a fundamental problem for learning sequential decision policies from observational data, as is necessarily the case in application areas such as education and healthcare Jiang & Li [11], Precup et al. [28], Thomas & Brunskill [36]. However, with a few exceptions, the literature on off-policy evaluation in reinforcement learning (RL) assumes (implicitly or otherwise) the *absence* of unobserved confounders, auxiliary state information that affects both the policy that generated the original data as well as transitions to the next state. Precisely in the same important domains where observational off-policy evaluation is *necessary* due to the cost of or ethical constraints on experimentation, such as in healthcare [27, 30] or operations, it is *also* generally the case that *unobserved confounders* are present. This contributes to fundamental challenges for advancing reinforcement learning in observational settings [7].

In this work, we study partial identification in RL off-policy evaluation under unobserved confounding, focusing specifically on the *infinite-horizon* setting. Recognizing that policy value cannot actually be point-identified from confounded observational data, we propose instead to compute the sharpest bounds on policy value that can be supported by the data and any assumptions on confounding. This can then support credible conclusions about policy value from the data and can ensure safety in downstream policy learning.

Recent advancements [6, 8, 13, 17] improve variance reduction of *unconfounded* off-policy evaluation by estimating density ratios on the *stationary* occupancy distribution. But this assumes unconfounded

---

[*]Corresponding author. Author ordering is alphabetical.

data. Other advances [14] tackle partial identification of policy values from confounded data but in the logged bandit setting (single decision point) rather than the RL setting (many or infinite decision points). Our work can be framed as appropriately combining these perspectives.

Our contributions are as follows: we establish a partially identified estimating equation that allows for the estimation of sharp bounds. We provide tractable approximations of the resulting difficult non-convex program based on non-convex first order methods. We then demonstrate the approach on a gridworld task with unobserved confounding.

## 2 Problem setup

We assume data is generated from an infinite-horizon MDP with an augmented state space: $\mathcal{S}$ is the space of the *observed* portion of the state and $\mathcal{U}$ is the space of the *unobserved* (confounding) portion of the state. We assume the standard decision protocol for MDPs on the full-information state space $\mathcal{S} \times \mathcal{U}$: at each decision epoch, the system occupies state $s_t, u_t$, the decision-maker receives a reward $\Phi(s_t)$ for being in state $s_t$ and chooses an action, $a_t$, from allowable actions. Then the system transitions to the next state on $\mathcal{S} \times \mathcal{U}$, with the (unknown) transition probability $p(s', u' \mid s, u, a)$. The full-information MDP is represented by the tuple $M = (\mathcal{S} \times \mathcal{U}, \mathcal{A}, P, \Phi)$. We let $\mathcal{H}_t = \{(s_0, u_0, a_0), \dots, (s_t, u_t, a_t)\}$ denote the (inaccessible) full-information history up to time $t$. A policy $\pi(a \mid s, u)$ is an assignment to the probability of taking action $a$ in state $(s, u)$. For any policy, the underlying dynamics are Markovian under full observation of states and transitions: $s_t \perp\!\!\!\perp \mathcal{H}_{t-2} \mid (s_{t-1}, u_{t-1}), a_{t-1}$.

In the off-policy evaluation setting, we consider the case where the observational data are generated under an unknown *behavior policy* $\pi_b$, while we are interested in evaluating the (known) *evaluation policy* $\pi_e$, which only depends on the observed state, $\pi_e(a \mid s, u) = \pi_e(a \mid s)$. Both policies are assumed stationary (time invariant). The observational dataset does not have full information and comprises solely of observed states and actions, that is, $(s_0, a_0), \dots, (s_t, a_t)$.[2] Thus, since the action also depends on the unobserved state $u_t$, we have that transition to next states are *confounded* by $u_t$.

Notationally, we reserve $s, u$ (respectively, $s', u'$) for the random variables representing state (respectively, next state) and we refer to realized observed state values (respectively, next observed state values) using $j$ (respectively, $k$). We assume that $\mathcal{S}$ is a discrete state space, while $\mathcal{U}$ may be general.

We next discuss regularity conditions on the MDP structure which ensure ergodicity and that the limiting state-action occupancy frequencies exist. We assume that the Markov chain induced by $\pi_e$ and any $\pi_b$ is a positive Harris chain, so the stationary distribution exists.

**Assumption 1** (Ergodic MDP). *The MDP $M$ is ergodic: the Markov chains induced by $\pi_b$ and $\pi_e$ is positive Harris recurrent.*

In this work, we focus on the infinite-horizon setting. Let $p_\pi^{(t)}(s)$ be the distribution of state $s_t$ when executing policy $\pi$, starting from initial state $s_0$ drawn from an initial distribution over states. Then the average *state-action-next-state* visitation distribution exists, and under Assumption 1 the (long-run average) *value* of a stationary policy $\pi_e$ is given by an expectation with respect to the marginalized state visitation distribution:

$$p_e^\infty(s, u, a, s', u') = \lim_{T \to \infty} \frac{1}{T} \sum_{t=0}^{T} p_e^{(t)}(s, u, a, s', u'), \qquad R_e = \mathbb{E}_{s \sim p_e^\infty}[\Phi(s)].$$

We similarly define the marginalized total-, unobserved-, and observed-state occupancy distributions as $p_\pi^\infty(s, u)$, $p_\pi^\infty(u)$, and $p_\pi^\infty(s)$, given by appropriately marginalizing the above. Notice we assumed that the reward only depends on the *observed* state[3].

Notationally, $\mathbb{E}$ denotes taking expectations over the joint stationary occupancy distribution of the behavior policy, where self-evident. We denote $p_e^\infty, p_b^\infty$ for visitation distributions induced under

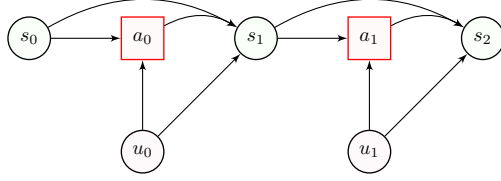

Figure 1: A causal model satisfying Assumption 2.

$\pi_e, \pi_b$. Since at times it is useful to distinguish between expectation over the marginalized occupancy distribution $p_b^\infty(s, a, s')$, and total expectation over full-information transitions $p_b^\infty(s, u, a, s', u')$, we include additional subscripts on the expectation whenever this is clarifying.

If we were able to actually run the MDP using the policy $\pi_e$, which is only a function of $s$, the dynamics would be Markovian with the marginalized transition probabilities:

$$ p(s' \mid s, a) := \sum_{u, u'} p(s', u' \mid s, u, a) p_e^\infty(u \mid s) $$

Note that $p(s' \mid s, a)$ is *not* identifiable from the observational data collected under $\pi_b$. We analogously define (partially) marginalized transition probabilities $p(s' \mid s, u, a)$.

## 3 Off-policy evaluation under unobserved confounding

In the following, we first discuss a *population* viewpoint, computing expectations with respect to the true marginalized stationary occupancy distribution $p_b^\infty(s, a, s')$ (identifiable from the data). We discuss finite-sample considerations in Section 6.

Given sample trajectories generated from $\pi_b$, the goal of off-policy evaluation is to estimate $R_e$, the value of a known (observed-state-dependent) evaluation policy $\pi_e(a \mid s)$. The full-information stationary density ratio is $w(s, u)$. If $w(s, u)$ *were* known, we could use it to estimate policy value under $\pi_e$ using samples from $\pi_b$ by a simple density ratio argument:

$$ w(s, u) = \frac{p_e^{(\infty)}(s, u)}{p_b^{(\infty)}(s, u)}, \qquad R_e = \mathbb{E}[w(s, u)\Phi(s)] $$

From observational data, we are only able to estimate the *marginalized* behavior policy, $\pi_b(a \mid s) = \mathbb{E}[\pi_b(a \mid s, u) \mid s]$, which is insufficient for identifying the policy value or the true marginalized transition probabilities.

### 3.1 Model restrictions on unobserved confounding

To make progress, we introduce restrictions on the underlying dynamics of the unobserved confounder, $u$, under which we will conduct our evaluation. In particular, we will seek to compute the range of all values of $R_e$ that match our data, encapsulated in $p_b^\infty(s, a, s')$, and the following structural assumptions.

**Assumption 2** (Memoryless Unobserved Confounding).

$$ \frac{p_e^{(\infty)}(s, u)}{p_b^{(\infty)}(s, u)} = \frac{p_e^{(\infty)}(s, u')}{p_b^{(\infty)}(s, u')} \qquad \forall s \in \mathcal{S}, u, u' \in \mathcal{U} $$

**Lemma 1.** *Assumption 2 holds if the MDP transitions satisfy*

$$ p(s', u' \mid s, u, a) = p(s', \tilde{u}' \mid s, u, a), \ \forall u', \tilde{u}' \in \mathcal{U}. $$

Assumption 2 essentially requires no time-varying confounders, *i.e.*, confounders that are influenced by past actions. This assumption may appear strong but it is necessary: if confounders could be time-varying and the dependence on them may be arbitrary, we may need to be "exponentially conservative" in accounting for them (or even "infinitely conservative" in the infinite-horizon case)[4].

For example, Assumption 2 holds if the unobserved confounder is exogenously drawn at each timestep, as in Figure 1, (which satisfies the sufficient condition of Lemma 1). Lemma 1 also holds if a confounded behavior policy arises from agents optimizing upon exogenous realizations of private information, as in econometric discrete choice models.

Under Assumption 2, we simply define $w(s) = w(s, u)$ as it does not depend on $u$. Note that $w(s)$ remains unidentifiable even under Assumption 2.

## 3.2 Sensitivity model

Next, we introduce a sensitivity model to control the level of assumed dependence of the behavior policy on the unobserved confounders. Sensitivity analysis allows a practitioner to assess how conclusions might change for reasonable ranges of $\Gamma$. Following [2, 14] we phrase this as lower and upper bounds on the (unknown) inverse behavior policy, $l(a \mid s), m(a \mid s)$:[5]

$$\beta(a \mid s, u) := \pi_b(a \mid s, u)^{-1}, \qquad l(a \mid s) \leq \beta(a \mid s, u) \leq m(a \mid s) \quad \forall a, s, u. \tag{1}$$

The set $\mathcal{B}$ consists of all functions $\beta(a \mid s, u)$ that satisfy Eq. (1). This ambiguity set is motivated by a sensitivity model used in causal inference, which restricts how far propensities can vary pointwise from the nominal propensities [34] and which has also been used in the logged bandit setting [14]. Given a sensitivity parameter $\Gamma \geq 1$ that controls the amounts of allowed confounding, the marginal sensitivity model posits the following odds-ratio restriction:

$$\Gamma^{-1} \leq \frac{(1 - \pi_b(a \mid s))\pi_b(a \mid s, u)}{\pi_b(a \mid s)(1 - \pi_b(a \mid s, u))} \leq \Gamma, \quad \forall a, s, u. \tag{2}$$

Eq. (2) is equivalent to saying that Eq. (1) holds with

$$l(a \mid s) = \Gamma/(\pi_b(a \mid s)) + 1 - \Gamma, \qquad m(a \mid s) = 1/(\Gamma \pi_b(a \mid s)) + 1 - 1/\Gamma.$$

Lastly, $\beta$ functions which are themselves valid inverse probability distributions must satisfy the next-state conditional constraints:

$$\mathbb{E}_{s,u,a,s' \sim p_b^\infty} \left[ \frac{\mathbb{I}[a = a']}{\pi_b(a' \mid s, u)} \mid s' = k \right] = p_b^\infty(k \mid a') \quad \forall k \in \mathcal{S}, a' \in \mathcal{A} \tag{3}$$

We let $\tilde{\mathcal{B}}$ denote the set of all functions $\beta(a \mid s, u)$ that satisfy *both* Eqs. (1) and (3).

## 3.3 The partially identified set

Given the above restrictions, we can define the set of partially identified evaluation policy values. To do so, suppose we are given a target behavior policy $\pi_e$, the observed stationary distribution $p_b^\infty(s, a, s')$, and bounds $l(a \mid s), m(a \mid s)$ on $\beta$. We are then concerned with what $w$ could be, given the degrees of freedom that remain. So, we define the following set for what values $w$ can take:[6]

$$\Theta = \left\{ \frac{p_e^{(\infty)}(s, u)}{p_b^{(\infty)}(s, u)} \quad : \quad \begin{array}{l} M \text{ is an MDP,} \\ M \text{ satisfies Assumptions 1 and 2 with respect to } \pi_b \text{ and the given } \pi_e, \\ \pi_b(a \mid s, u) \text{ is a stationary policy with } \beta \in \mathcal{B} \text{ and } p_b^\infty(s, a, s') \text{ as given} \end{array} \right\}$$

We are then interested in determining the largest and smallest that $R_e$ can be. That is, we are interested in

$$\underline{R}_e = \inf_{w \in \Theta} \mathbb{E}[w(s)\Phi(s)], \quad \overline{R}_e = \sup_{w \in \Theta} \mathbb{E}[w(s)\Phi(s)]. \tag{4}$$

Notice that this is equivalent to computing the *support function* of $\Theta$ at $-\Phi$ and $\Phi$ with respect to the $L_2$ inner product defined by $p_b^\infty(s)$, $\langle f, g \rangle = \mathbb{E}[f(s)g(s)] = \sum_j p_b^\infty(j)f(j)g(j)$. The support function of a set $\mathcal{S}$ is $\psi(v) = \sup_{s \in \mathcal{S}} \langle v, s \rangle$ [32].

# 4 Characterizing the partially identified set

In this section we derive a linear program to check membership in $\Theta$ for any given $w$.

**The partially identified estimating equation** We begin by showing that $w$ is uniquely characterized by an estimating equation characterizing its stationarity, but where some parts of the equation are not actually known.

**Lemma 2.** *Suppose Assumptions 1 and 2 hold. Then* $w(s) = \frac{p_e^{(\infty)}(s,u)}{p_b^{(\infty)}(s,u)} \; \forall s, u$ *if and only if*

$$\mathbb{E}[\pi_e(a \mid s)w(s)\beta(a \mid s, u) \mid s' = k] = w(k) \;\; \forall k, \tag{5}$$

$$\mathbb{E}[w(s)] = 1. \tag{6}$$

The forward implication of Lemma 2 follows from Theorem 1 of [17] applied to the state variable $(s, u)$ after recognizing that $w(s, u)$ only depends on $s$ under Assumption 2 and marginalizing out $u'$. The backward implications of Lemma 2 follows from the recurrence of the aggregated MDP obtained from the transform $(s, u) \mapsto s$ [16]. A complete proof appears in the appendix.

Fortunately, Eqs. (5) and (6) exactly characterize $w$. Unfortunately, Eq. (5) involves two unknowns: $\beta(a \mid s, u)$ and the distribution $p_b^\infty(s, u, a, s')$ with respect to which the expectation is taken. In that sense, the estimated equation is only partially identified. Nonetheless, this allows to make progress toward a tractable characterization of $\Theta$.

**Marginalization** We next show that when optimizing over $\beta \in \mathcal{B}$, the sensitivity model can be reparametrized with respect to *marginal weights* $g_k(a \mid j)$ (in the following, $j, a, k$ are generic indices into $\mathcal{S}, \mathcal{A}, \mathcal{S}$, respectively):

$$g_{s'}(a \mid s) := \sum_{\tilde{u}} \frac{p_b^{(\infty)}(s, \tilde{u}, a \mid s')}{p_b^\infty(s, a \mid s')}\beta(a \mid j, \tilde{u}) = \left(\sum_{\tilde{u}} \pi_b(a \mid j, \tilde{u}) \frac{p_b^\infty(\tilde{u} \mid s)p(s' \mid s, \tilde{u}, a)}{p(s' \mid s, a)}\right)^{-1}$$

Note that the values of the $g_{s'}(a \mid s)$ weights are *not* equivalent to the confounded $\pi_b(a \mid s)^{-1}$: the difference is exactly the *variability* in the underlying full-information transition probabilities $p(s' \mid s, u, a)$. We will show that $g_{s'}(a \mid s) \in \tilde{\mathcal{B}}$ satisfies the following constraints, where Eq. (7) corresponds to Eq. (3):

$$l(a \mid s) \le g_{s'}(a \mid s) \le m(a \mid s), \qquad\qquad \forall s, s' \in \mathcal{S}, a \in \mathcal{A}$$

$$p_b^\infty(s' \mid a) = \sum_{s \in \mathcal{S}} p_b^\infty(s, a, s')g_{s'}(a \mid s), \qquad\qquad \forall s' \in \mathcal{S}, a \in \mathcal{A} \tag{7}$$

Reparametrization with respect to $g_{s'}(a \mid s)$ follows from an optimization argument, recognizing the symmetry of optimizing a function of unknown realizations of $u$ with respect to an unknown conditional visitation density. Crucially, reparametrization improves the scaling of the number of nonconvex bilinear variables from the number of samples or trajectories, $O(NT)$, to $O(|\mathcal{S}|^2|\mathcal{A}|)$.

Unlike sensitivity models in causal inference, it is possible that the partial identification set is empty, $\Theta = \emptyset$, even if its associated sensitivity model $\tilde{\mathcal{B}}$ is nonempty in the space of weights. In Appendix B.1 of the appendix, we explain this further by studying the state-action-state polytope [20], the set of all valid visitation distributions achievable by some policy. The next result summarizes that imposing constraints on the *marginalized* joint distribution $p_b^\infty(s, a, s')$ is insufficient to ensure the full (unobserved) joint distribution corresponds to a valid MDP stationary distribution.

**Proposition 1.** *The implementable implications of the marginalized state-action and marginalized state-action-state polytopes are:* $p_b^\infty(s' \mid a) = \sum_s p_b^\infty(s, a, s')g_{s'}(a \mid s), \forall s, s' \in \mathcal{S}, a \in \mathcal{A}.$

Proposition 1 justifies our restrictions on $\tilde{\mathcal{B}}$. However, it also implies that further imposing constraints on $\tilde{\mathcal{B}}$ *cannot* ensure compatibility of $g_{s'}(a \mid s)$ for the observed $p_b^\infty(s, a, s')$ (and therefore $\Theta$), where compatibility is the requirement that $p_b^\infty(s, a, s')$ is stationary for $g_{s'}(a \mid s)$.

**Feasibility Linear Program** We next show that $w \in \Theta$ can be expressed using the linear program $F(w)$ that minimizes the L1 norm of residuals of the estimating equation of Lemma 2, for a given $w$, over the sensitivity model $g \in \tilde{\mathcal{B}}$[7]:

$$F(w) := \min_{g \in \tilde{\mathcal{B}}} \sum_{s' \in \mathcal{S}} \left| \sum_{s \in \mathcal{S}, a \in \mathcal{A}} p_b^{(\infty)}(s, a \mid s')w(s)\pi_e(a \mid s)g_{s'}(a \mid s) - w(s') \right|. \tag{8}$$

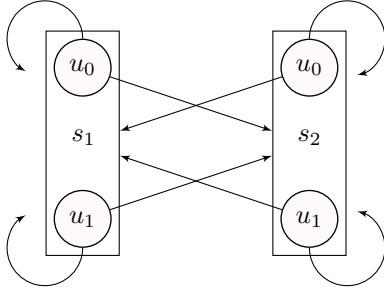

Figure 2: Confounded random walk.

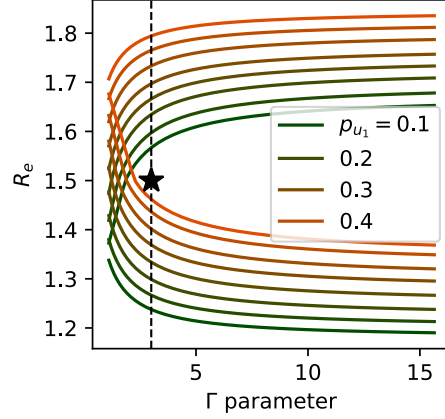

Figure 3: Varying transition models on $(s, u)$. $\star$ is true policy value.

**Proposition 2** (Feasibility Linear Program).

$$w \in \Theta \iff F(w) \leq 0, \ \mathbb{E}[w(s)] = 1 \tag{9}$$

In relation to Proposition 1, the analysis for Eq. (9) shows that it is exactly the partially identified estimating equation, Lemma 2, which enforces compatibility such that combining the restrictions on $\tilde{\mathcal{B}}$ and Lemma 2 verifies membership of $w$ in $\Theta$. A consequence of Proposition 2 is sharpness of the partially identified interval $[\underline{R}_e, \overline{R}_e]$; each point in the interval corresponds to some policy value.

**Theorem 1** (Sharpness).

$$\{\mathbb{E}[w(s)\Phi(s)] : w \in \Theta\} = [\underline{R}_e, \overline{R}_e].$$

## 5 Optimizing over the partially identified set

Eq. (9) suggests computing $\underline{R}_e, \overline{R}_e$ by solving

$$\inf / \sup \ \{\mathbb{E}[w(s)\Phi(s)] : F(w) \leq 0, \ \mathbb{E}[w(s)] = 1\}. \tag{10}$$

The restriction $F(w) = 0$ implicitly encodes an optimization over $g$, resulting in a hard nonconvex optimization problem. We outline some computational approaches.

**Global optimization** Computationally, global optimization is possible via branch and bound approaches for the nonconvex quadratic program obtained by solving Equation (10) directly, simultaneously over $w, g$. We take this approach in the experiments, where we use Gurobi version 9. However, this approach is not scalable for large or moderately sized state spaces.

**Disjunctive reformulation** We show that the structure of the feasibility program admits reformulation as a disjunctive program.

**Proposition 3.** *Eq. (10) can be reformulated as a disjunctive program (hence a finite linear program).*

However, the size of the resulting program is super-exponential in the cardinality of the state space, hence not practical. In Appendix E we also discuss the computational intractability of lifted SDP relaxations, which incur $O(|\mathcal{S}|^9)$ iteration complexity. Therefore, for small state spaces, we suggest to solve Eq. (10) directly via Gurobi.

**Nonconvex nonconvex-projected gradient method** We next develop a more practical optimization approach based on non-convex first-order methods. First we restate the estimating equation Eq. (5) for a fixed $g$ as a matrix system. To evaluate expectations on the unconditional joint distribution, we introduce instrument functions $\phi_s, \phi_{s'} \in \mathbb{R}^{|\mathcal{S}| \times 1}$, random (row) vectors which are one-hot indicators for the state random variable $s, s'$ taking on each value, $\phi_s =$

---

**Algorithm 1** Nonconvex nonconvex-projected gradient descent

---

**Input:** step size $\eta_0$, exponent $\kappa \in (0, 1]$, initial iterate $g_0$, number of iterations $N$
**for** $k = 0, \ldots, N - 1$ **do**
$\quad \eta_k \leftarrow \eta_0 t^{-k}$
$\quad w_{g_k}^* \in \arg\min_{w \geq 0}\{\|A(g_k)w\|_1, \mathbb{E}[w(s)] = 1\}, \tilde{g}_k \in \arg\min_{g \in \tilde{\mathcal{B}}}\{\|g - g_k\|_1 : A(g)w_{g_k}^* = 0\}$
$\quad g_{k+1} \leftarrow \text{Proj}_{\mathcal{B}}(g_k + \eta_t \nabla_{\tilde{g}_k}(\varphi^T \tilde{A}(g)^{-1}v))$
**end for** $\quad$ Return $g_k$ with the best loss.

---

$[\mathbb{I}[s = 0] \quad \ldots \quad \mathbb{I}[s = |\mathcal{S}|]]$. Let $A(g) = \mathbb{E}[\phi_{s'}(\pi_e(a \mid s)g_{s'}(a \mid s)\phi_s - \phi_{s'})^\top]$ and $b_s = p_b^\infty(s)$. Let $\psi$ be the set of $g \in \tilde{\mathcal{B}}$ that admit a feasible solution to the estimating equation for some $w \in \Theta$:

$$\psi := \{g \in \tilde{\mathcal{B}} : \exists\, w \geq 0 \text{ s.t. } A(g)w = 0, b^\top w = 1\} \tag{11}$$

Define $\tilde{A}(g)$ by replacing the last row of $A(g)$ by $b$ and let $v = (0, \ldots, 0, 1) \in \mathbb{R}^{|\mathcal{S}|}$.

**Proposition 4.** *If $g \in \psi$ then $\tilde{A}(g)$ is invertible. Moreover, $\Theta = \{\tilde{A}(g)^{-1}v : g \in \psi\}$.*

Proposition 4 suggests computing $\underline{R}_e, \overline{R}_e$ by solving

$$\inf / \sup \{\varphi^T \tilde{A}(g)^{-1}v : g \in \psi\}, \tag{12}$$

where $\varphi_s = \Phi(s)p_b^\infty(s)$. This optimization problem has both a non-convex objective and a non-convex feasible set, but it has small size. As a way to approximate $\underline{R}_e, \overline{R}_e$, we propose a gradient descent approach to solving Eq. (12) in Algorithm 1. Since the feasible set is itself non-convex, we use an approximate projection that corrects each $g_k$ variable at iteration $k$ to a feasible point but may not be a projection. This is based on taking alternating projection steps on $w_{g_k}^* \in \arg\min\{\|A(g_k)w\|_1 : \mathbb{E}[w(s)] = 1\}$ and $\tilde{g}_k \in \arg\min\{\|g - g_k\|_1 : A(g)w_{g_k}^* = 0, g \in \tilde{\mathcal{B}}\}$; each of these is a linear program with $|\mathcal{S}|$ or $|\mathcal{S}|^2|\mathcal{A}|$ many variables, respectively.

**Remark 2.** Since the tabular setting is a special case of linear function approximation for $w$, our approach directly handles the case where $w = \theta^\top s$ is a linear function of the state, but further requires well-specification. See Appendix D.1 for detail and discussion of additional challenges.

## 6 Consistency

The above analysis considered the population setting, but in practice we use the empirical state-action occupancy distribution, $\hat{p}_b^\infty(s, a, s')$. Define $\hat{\overline{R}}_e, \hat{\underline{R}}_e$ as the corresponding values when we solve Eq. (4) with this estimate in place of $p_b^\infty(s, a, s')$. We establish consistency of the estimated bounds.

**Theorem 3** (Consistency). *If $\hat{p}_b^\infty(s, a, s') \to p_b^\infty(s, a, s')$, then*

$$\hat{\overline{R}}_e \to \overline{R}_e, \hat{\underline{R}}_e \to \underline{R}_e.$$

Since the empirical distributions satisfy $\hat{p}_b^\infty(s, a, s') \to_p p_b^\infty(s, a, s')$ [see, *e.g.*, 20, Theorem 3] and the continuous mapping theorem would then together imply that $\hat{\overline{R}}_e \to_p \overline{R}_e$, $\hat{\underline{R}}_e \to_p \underline{R}_e$. Since the perturbation of $p_b^\infty$ to $\hat{p}_b^\infty$ introduces perturbations to the *constraint matrix* of the LP, to prove this result we leverage a general stability analysis due to [31, Theorem 1].

## 7 Empirics

**Illustrative example: Confounded random walk** We introduce a simple example in Figure 2. Figure 2 satisfies a sufficient condition of Lemma 1 this is graphically denoted by arrows from $(s, u)$ tuples to $s'$ states. The confounded random walk is parametrized by the transition probabilities under action $a = 1$: $p(s_1 \mid s_1, u_1, a = 1) = p_{u_1}$, $p(s_1 \mid s_1, u_2, 1) = \frac{1}{2} - p_{u_2}$, where $u$ is generated exogenously upon transiting a state $s$. Transitions are antisymmetric under action $a = 2$, so that $p(s_1 \mid s_1, u_1, 1) = \frac{1}{2} - p_{u_1}$, $p(s_2 \mid s_1, u_2, 1) = p_{u_2}$. Then, a stationary policy that is uniform over actions generates a random walk on $\mathcal{S}$. See Appendix E for a full description.

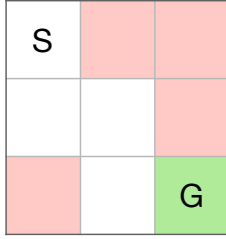

Figure 4: 3x3 gridworld.
$\Phi(s)$ of green is 1, red -0.3.

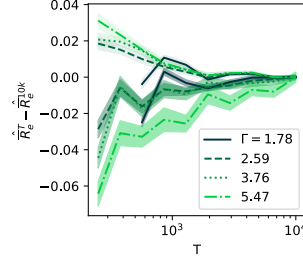

Figure 5:
Statistical consistency

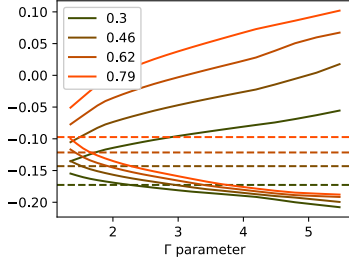

Figure 6: Bounds when $\pi_b$ is $\eta-$weighted mixture of $\pi_b^{*,u=0}$, the $(s,0)$-optimal policy, and uniform.

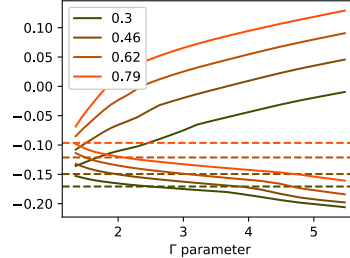

Figure 7: $\pi_b$ is mixture policy of $\pi_b^{*,\mathcal{S}}$, $\mathcal{S}$ optimal policy, to uniform.

In Fig. 3, we vary the underlying transition model, varying $p_{u_1} = p_{u_2}$ on a grid $[0.1, 0.45]$, and we plot the varying bounds with action-marginal control variates. We compute bounds via global optimization with Gurobi version 9 (each problem solves within seconds). The true underlying behavior policy takes action $a = 1$ with probability $\pi(1 \mid s_1, u_1) = \pi(1 \mid s_2, u_1) = \frac{1}{4}$ (and the complementary probability when $u = u_2$), modeling the setting where a full-information behavior policy is correlated with the full-information transitions. The behavior policy on $\mathcal{S}$ appears to be uniform; using these confounded estimates results in biased estimates of the transition probabilities. As we vary the transition model in Fig. 3, note that the true policy value when $\Phi = [1, 2]$ is 1.5, indicated by $\star$, and it is within the bounds for large enough $\Gamma = 3$, uniformly over different data-generating processes.

**3x3 confounded windy gridworld.** We introduce unobserved confounding to a 3x3 simple windy gridworld environment, depicted in Figure 4 in the appendix [33]. The the agent receives $\Phi(s) = 1$ reward at a goal state but $\Phi(s) = -0.3$ at hazard states (shaded red). We assume a binary unobserved confounder $u \in \{0, 1\}$ that represents "wind strength". Transitions in the action direction succeed in that direction with probability $p = 0.8$, otherwise with probability $0.1$ the agent goes east or west. However, when $u = 1$, the "westward wind" is strong, but if the agent takes action "east", the agent instead stays in place (otherwise the agent transitions west). The wind is generated exogenously from all else in the environment. An optimal full-information behavior policy (agent with wind sensors) varies depending on $u_t$ by taking the left action, to avoid the penalty states. This models the setting where unobserved confounding arises due to exogenous private information.

In Figure 5, we study the finite-sample properties of the bounds estimator, plotting $\overline{\hat{R^T}}_e - \overline{\hat{R_e^{10k}}}$ for differing trajectory lengths on a logarithmic grid, $T \in [250, 10000]$, and standard errors averaged over 50 replications. (We plot the difference in order to normalize by an estimate of the large-sample limit). We compute bounds via global optimization with Gurobi version 9 (each problem solves within seconds). The bounds converge; optimizing over larger $\Gamma$ values tends to increase finite sample bias.

We illustrate bounds obtained by our approach in Figure 6 for evaluation policies which are mixtures to $\pi_b^{*,u=0}$, a suboptimal policy that is optimal for the transitions when $u = 0$ (no wind), and in Figure 7, $\pi^{*\mathcal{S}}$, a policy that is optimal on $\mathcal{S}$ given the true marginalized transition probabilities (which

are unknown to the analyst, but known in this environment). We display the bounds as we range mixture weights on the non-uniform policy from $0.3$ to $0.8$. We display in a dashed line, with the same color for corresponding mixture weight $\eta$, the true value of $R_e$.

# 8   Related work, discussion, and conclusions

**Off-policy evaluation in RL.**   We build most directly on a line of recent work in off-policy policy evaluation which targets estimation of the stationary distribution density ratio [6, 8, 13, 17], which can be highly advantageous for variance reduction compared to stepwise importance sampling.

**Sensitivity analysis in the batch setting in causal inference.**   Sensitivity analysis is a rich area in causal inference. A related work in approach is [14], which builds on [2] and considers robust off-policy evaluation and learning in the *one-decision-point* setting. In Appendix A.1, we discuss an extension of their or other inverse-weight robust approaches to this setting and the inherent challenges of such an approach with long horizons in introducing "exponential robustness". The identification approach in this work is very different: the partial identification region is only identified *implicitly* as the solution region of an estimating equation. Unlike [14], [4] consider an *identifiable* setting where we are given a well-specified latent variable model and propose a minimax balancing solution. Finally, [15, 39] study bounds for conditional average treatment effects in the *one-decision-point* setting. [40] consider sensitivity analysis under assumptions on the outcome functions for a marginal structural model.

**Off-policy evaluation in RL with unobservables.**   Various recent work considers unobserved confounders in RL. [23] considers identification of counterfactuals of trajectories in an POMDP and SCM model. [35] study off-policy evaluation in the POMDP setting, proposing a "decoupled POMDP" and leveraging the identification result of [21], viewing previous and future states as negative controls. [19] propose a "deconfounded RL method" that builds on the deep latent variable approach of [18]. [41] uses partial identification bounds to narrow confidence regions on the transition matrix to warm start the UCRL algorithm of [10].

These generally consider a setting with sufficient assumptions or data to render policy values *identifiable*, where in the general observational setting they are unidentifiable. Specifically, [35] require an invertibility assumption that implies in a sense that we have a proxy observation for *every* unobserved confounder, [19] assume a well-specified latent variable model, also requiring that every unobserved confounder is reflected in the proxies, and [41] consider an online setting where additional experimentation can eventually identify policy value. Our approach is complementary to these works: we focus on the time-homogeneous, infinite-horizon case, and are agnostic to distributional or support assumptions on $u$. Our structural assumption on $u$'s dynamics (Assumption 2) is also new.

In contrast to POMDPs in general, which *emphasize* the hidden underlying state and its statistical recovery, our model is distinct in that we focus on rewards as functions of the observed state. In contrast to robust MDPs, one may derive a corresponding ambiguity set on transition matrices (see Lemma 4 in the appendix), but it is generally *non-rectangular* because the ambiguity set does not decompose as a product set over states, which leads to a NP-hard problem in the general case [38]. See Appendix A.2 of the appendix for a fuller discussion.

**Conclusions and future work**   Our work establishes partial identification results for policy evaluation in infinite-horizon RL under unobserved confounding. We showed that the set of all policy values that agree with both the data and a marginal sensitivity model can be expressed in terms of whether the value of a linear program is non-positive and developed algorithms that assess the minimal- and maximal-possible values of a policy. Further algorithmic improvements are necessary in order to extend our results to infinite state spaces. Finally, an important next step is to translate the partial identification bounds to robust policy learning.

**Funding disclosure**  This material is based upon work supported by the National Science Foundation under Grants No. 1939704 and 1846210.

## Broader Impact

As a contribution to offline RL, our work is of particular importance for RL in the context of social and medical sciences, where experimentation is limited and observational data must be used. The validity of the no-unobserved-confounders assumption is of particular importance to applied research in these fields since the presence of unobserved confounding can bias standard evaluations that assume the issue away and, unchecked, this may potentially hide harms done by the policy being evaluated or learned. Our work is the first step in developing offline RL algorithms that directly address this real, practical issue, and its primary purpose is to directly deal with such biases in the data.

That said, it is well understood that there is generally a gap between theory and practice in RL as it is applied to very complex and large-scale systems. It is therefore important to keep in mind practical heuristics, stopgaps, and approximations from applied RL when translating this work into practice. The systematic investigation of the use of these in the context of confounded offline RL and more extensive experimental studies in larger environments may require additional future work.

Moreover, there are several general potential dangers to be cognizant of when applying any offline RL tool in practice. First, if the observational data is not representative of the population, that is, there is a covariate shift in the state distribution, then the evaluation will reflect these biases and correspondingly be unrepresentative, under-emphasizing value to some parts of the population and over-emphasizing value to others. More generally, even without covariate shift, here we focused on evaluation of *average* welfare, which may average the harms to some and the benefits to others; it may therefore be important in some applications to also conduct auxiliary evaluations on certain protected subgroups to ensure equal impact, which can be done by segmenting the data. Third, it is possible that any offline RL approach may be applied inappropriately when the assumptions are not met – here we dealt directly with how to deal with violations of unconfoundedness assumptions but there may still be other assumptions such as our Assumption 1 – and this will mislead any evaluation or learning. For example, concerns about violations of absolute continuity of importance weights (overlap) are especially relevant in the offline RL setting. Therefore, we strongly recommend considering such offline RL and, in particular the sensitivity analyses we developed herein, as a way to inform further investigation, additional data collection, and/or investment in a randomized trial, rather than as an outright replacement for any of these. That said, offline evaluation, especially robust evaluation as we propose herein, is crucial for assessing policies *before* even trialling them, where they may effect actual negative impacts on study populations.

## Footnotes

[2]Our model differs from typical POMDPs [12], since rewards are a function of observed state, as we clarify in the related work, Section 8.

[3]This does not preclude, however, dependence on action: if we are given observed-state-action reward function $\Phi'(s, a)$, we may simply define $\Phi(s) = \sum_a \pi_e(a \mid s) \Phi'(s, a)$, since $\pi_e(a \mid s)$ is assumed known. Then $R_e$ gives $\pi_e$'s value with respect to the given observed-state-action reward function.

[4]We discuss this in Appendix A.1 of the appendix in considering a finite-horizon off-policy estimate.

[5]Our approach immediately generalizes to any linearly representable ambiguity set.

[6]Note that, as defined, $\Theta$ is a set of functions of $(s, u)$ but because we enforce Assumption 2, all members are constant with respect to $u$ for each $s$; we therefore often implicitly refer to it as a set of functions of $s$ alone.

[7]Linearity of $F(w)$ also holds if an instrument function is used to convert the conditional moment equality to an unconditional one, as in Eqn. 10 [17], and as we use in Section 5 and Proposition 4.

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
