[Supplementary Material · neurips_supplementary.pdf]

**Outline of the appendix**

- Appendix A contains omitted discussion and comparison to other frameworks.

- Appendix B contains proofs regarding the model and partial identification thereof.

- Appendix C discusses optimization and algorithms.

- Appendix D proves statistical consistency.

- Appendix E contains additional empirics and computational discussion.

## A  Additional context

### A.1  Off Policy Policy Evaluation: Relationship to finite-horizon case

To aid comparison to the off-policy policy evaluation literature, we describe an approach to robustness under unobserved confounding which might be pursued in the finite-horizon case, which does not leverage stationarity. Such an approach would bound the density ratio product of true behavior policy weights $\prod_{t \in [H]} \frac{\pi_e(a_t|s_t)}{\pi_b(a_t|s_t,u_t)}$ relative to the product of nominal inverse propensity weights, $\prod_{t \in [H]} \frac{\pi_e(a_t|s_t)}{\pi_b(a_t|s_t)}$ (including the moment restrictions according to it being a valid density ratio). It is apparent by factorizing the joint distribution that the true density ratio product would identify the policy value.

$$
\begin{aligned}
R_e =& \mathbb{E}[\sum_{h=1}^{H} r_h \mid a_{1:H} \sim \pi_e] \\
=& \mathbb{E}_b \left[ \left( \frac{p^{(0)}(s_0, u_0)}{p^{(0)}(s_0, u_0)} \prod_{t \in [H]} \frac{\pi_e(a_t \mid s_t) p(r_t \mid s_t, a_t) p(s_{t+1}, u_{t+1} \mid s_t, u_t, a_t)}{\pi_b(a_t \mid s_t, u_t) p(r_t \mid \mathcal{H}_t) p(s_{t+1}, u_{t+1} \mid s_t, u_t, a_t)} \right) \sum_{h=1}^{H} r_h \right] \\
=& \mathbb{E}_b \left[ \prod_{t \in [H]} \frac{\pi_e(a_t \mid s_t)}{\pi_b(a_t \mid s_t, u_t)} \sum_{h=1}^{H} r_h \right]
\end{aligned}
$$

While optimizing bounds on the range of the product of importance-sampling weights, $\prod_{t \in [H]} \frac{\pi_e(a_t|s_t)}{\pi_b(a_t|s_t,u_t)}$ may be tractable using geometric programming [5], enforcing the moment restrictions on density ratios (as in Eq. (7)) introduces further difficulty (nonconvex equality constraints). The core difficulty is that considering an uncertainty set which decomposes as a product set over timesteps may be too conservative to be useful in practice.

Further, in the infinite horizon setting, the divergence of the robust density ratio products grows with the horizon length. This is immediate when considering a relaxation of the sharp uncertainty set, i.e. optimizing the density ratio within bounds without enforcing moment constraints that must be satisfied by inverse probabilities by a geometric programming reformulation [5]; verifying similar properties in the sharp setting with additional nonlinear constraints may require additional analysis.

### A.2  Related work and relation to POMDPs and robust MDPs

**Contrast to POMDPs.** In contrast to POMDPs in general, which *emphasize* the hidden underlying state, our model is distinct in that we focus on rewards as functions of the observed state. The unobserved confounder is therefore a "*nuisance*" confounder which prevents us from estimating policy value, rather than the true underlying state to recover. In settings where unobserved confounders are of concern in observational data in causal inference, typically it is unclear whether or not a latent variable model such as those underlying POMDPs indeed generalizes to the time of deployment: assuming so corresponds to a structural assumption about the environment.

**Contrast to Robust MDPs.** Robust MDPs, representing a *model-based* approach, consider policy evaluation or improvement over an ambiguity set of the *transition probabilities* [9, 22, 38].

Alternatively, some approaches build confidence regions from concentration inequalities [26, 37] and restrict recommendations within them. [25] improve performance guarantee bounds for state aggregation in MDPs; but in their setting they are able to sample additional full-information transitions unlike our fully-observational data setting The difficulty in applying the robust MDP framework using an ambiguity set on transition matrices suggested from Lemma 4 (in the appendix) is *non-rectangularity* because the ambiguity set does not decompose as a product set over states, which leads to a NP-hard problem in the general case [38].

## B   Proofs: Model and Partial Identification

*Proof of Lemma 1.* This is apparent from a recursive definition of $w(j, u)$:

$$
\begin{aligned}
w(s, u) &= \lim_{T \to \infty} \frac{\sum_{t=1}^{T} \sum_{a,k,u'} p(j, u \mid k, u', a) p_e^{t-1}(k, u', a)}{\sum_{n=1}^{T} \sum_{a,k,u'} p(j, u \mid k, u', a) p_b^{t-1}(k, u', a)} \\
&= \lim_{T \to \infty} \frac{\sum_{t=1}^{T} \sum_{a,k,u'} p(j, u'' \mid k, u', a) p_e^{t-1}(k, u', a)}{\sum_{n=1}^{T} \sum_{a,k,u'} p(j, u'' \mid k, u', a) p_b^{t-1}(k, u', a)} = w(s, u'')
\end{aligned}
$$

$\square$

*Proof of Lemma 2.* Clearly, by assumption of Markovian dynamics on the full information state space, $w(s, u)$ solves the estimating equation (state-action flow equations) on the space of $\mathcal{S} \times \mathcal{U}$,

$$
\mathbb{E}[\pi_e(a \mid s) w(s, u) \beta(a \mid s, u) \mid s' = k] = w(k, v) \ \ \forall k \in \mathcal{S}, v \in \mathcal{U} \tag{13}
$$
$$
\mathbb{E}[w(s, u)] = 1. \tag{14}
$$

We will proceed to show that under Assumption 2, the projected $\tilde{w}(k)$ defined as $\tilde{w}(k) := w(k, v)$ equivalently solves the estimating equation on the observed state space $\mathcal{S}$.

The forward implication of Lemma 2 follows from Theorem 1 of Liu et al. [17] applied to the state variable $(s, u)$ after recognizing that $w(s, u)$ only depends on $s$ under Assumption 2 and marginalizing out $u'$.

$$
\begin{aligned}
\tilde{w}(k) = \frac{p_e^{\infty}(k)}{p_b^{\infty}(k)} = w(k, v) &= \mathbb{E}_{(s,u),a,(s',u') \sim p_b^{\infty}}[w(s, u) \pi_e(a \mid s) \beta(a \mid s, u) \mid s' = k, u' = v] \\
&= \mathbb{E}_{(s,u),a,s' \sim p_b^{\infty}}[w(s, u) \pi_e(a \mid s) \beta(a \mid s, u) \mid s' = k] \qquad \text{by Assumption 2} \\
&= \mathbb{E}_{(s,u),a,s' \sim p_b^{\infty}}[\tilde{w}(s) \pi_e(a \mid s) \beta(a \mid s, u) \mid s' = k]
\end{aligned}
$$

Verifying that $\tilde{w}(k) p_b^{(\infty)}(k)$ satisfies conditions on the invariant measure on $s$:

$$
\begin{aligned}
\tilde{w}(k) &= \frac{1}{p_b^{(\infty)}(k)} \sum_{j,a} \tilde{w}(j) \sum_u p(k \mid j, u, a) p_b^{(\infty)}(s, u) \pi_b(a \mid s, u) \frac{\pi(a \mid s)}{\pi_b(a \mid s, u)}, \forall k \\
&= \frac{1}{p_b^{(\infty)}(k)} \sum_{j,a} \tilde{w}(j) \pi(a \mid j) \sum_u p_b^{(\infty)}(j, u) p(k \mid j, u, a), \forall k
\end{aligned}
$$

Therefore,

$$
\tilde{w}(k) p_b^{(\infty)}(k) = \sum_{j,a} \tilde{w}(j) p_b^{(\infty)}(j) \pi(a \mid j) p(k \mid j, a),
$$

so we conclude that $\tilde{w}(s) \propto p_e^{(\infty)}$, the stationary distribution on $\mathcal{S}$ induced by $\pi_e$.

Finally, we argue the reverse implication; uniqueness of the solution of $\tilde{w}(s)$. Uniqueness is a consequence of the positive recurrence assumption (Assumption 1) on the full-information MDP on $\mathcal{S} \times \mathcal{U}$. Note that by definition of recurrence, recurrence on the full-observation state space of the Markov process induced under $\pi$ implies recurrence of the Markov process induced under $\pi$ on

its marginalized transitions $p(k \mid j, a)$. Recurrence requires that starting from any state $j, u$ in the recurrent class, the number of visits of the chain to the state is infinite. Clearly, if this is satisfied by the full-information transition matrix, this is also satisfied for the aggregated recurrent class corresponding to marginalized transitions.

Therefore, the stationary distribution exists, and is unique on $\mathcal{S}$, under the marginalized transition matrix induced by $\pi_e$. The solution to the invariant measure flow equations on $\mathcal{S}$ satisfies that $\tilde{w}(s)p_b^\infty(s) \propto p_e^\infty(s)$; and only $\tilde{w}(s)$ satisfies this requirement. $\qquad\square$

## B.1 Observable Implications, Membership Oracle, and Sharpness

In this section, we introduce the state-action polytope and the state-action-state frequency polytope and deduce the observable implications which lead to Proposition 1, the main membership certificate result of Proposition 2, and Theorem 1.

For any infinite-horizon MDP with transition matrix on $\mathcal{S} \times \mathcal{U}$, the stationary dynamics impose restrictions on the unknown full-information state-action-state visitation distribution, $p_b^\infty(s, u, a, s', u')$, and its observable marginalization $p_b^\infty(s, a, s')$. These restrictions are encapsulated as the full-information *state-action polytope* (SAP), which is the set of all limiting state-action occupancy probabilities achievable under any policy, and the closely related *state-action-state* polytope (SASP) [20, 29]. Marginalizing the full-information constraints with respect to $p_b^\infty(u \mid s)$ leads to the marginalized versions mSAP and SASP.

*Proof of Proposition 1.* To study the observational implications of SAP, SASP (e.g. the implications of the full-information polytopes which are additionally enforceable as constraints on $\tilde{\mathcal{B}}$), we study the marginalized versions of both the state-action polytope and the state-action-state frequency polytope under the behavior policy as studied in [20, 29]. Typically the extremal analysis of the state-action polytope in the infinite-horizon case characterizes the structure of the optimal policy. We simply focus on its properties as a characterization of all the possible limiting state-action frequencies under *any* stationary policy.

While [20] studies the asymptotic inclusion of more general policies such as non-stationary policies, we focus on the case of stationary policies for simplicity. While we state the following analysis for discrete state and action spaces (e.g. discrete $u$), the discussion of Altman & Shwartz [1, Sec. 4] provides regularity results on the set of limiting state-action measures for the continuous state case with continuous $u$. One sufficient condition for the case of continuous $u$ is that, indexing $x, y$ into abstract state tuples on $\mathcal{S} \times \mathcal{U}$, for the transition probability density defined as $P_{x,K}^\pi := \sum_{y \in K} P_{x,y}^\pi$, with $P_{x,y}^\pi = \mathbb{P}(s_{t+1} = y \mid s_t = x)$, given any $\epsilon > 0$ there exist a finite set $K(\epsilon)$ and an integer $N(\epsilon)$ such that for all $x \in X$ and $g \in U(S)$, $[(P^\pi)^{N(\epsilon)}]_{x,K(\epsilon)} \geq 1 - \epsilon$.

First, we introduce SAP, SASP as studied in [20].

**Definition 1** (State-action polytope). Given an MDP, the **state-action polytope** SAP is defined as the set of vectors $x$ in $\Delta^{\mathcal{S} \times \mathcal{A}}$ that satisfy

$$\sum_{j,u'} \sum_a p(k, u'' \mid j, u', a)p_b^\infty(j, u', a) = \sum_{a'} p_b^\infty(k, u'', a'), \qquad \forall s' \tag{15}$$

where $p_b^\infty(j, u', a) \in \Delta^{\mathcal{S} \times \mathcal{A}}$ is the limiting expected state-action frequency vector under policy $\pi$. This constraint can be understood as a "flow conservation" constraint satisfied by any full-information joint distribution $p_b^\infty(s, u, a, s', u')$.

**Definition 2** (State-action-state polytope). The **state-action-state** frequency polytope, SASP, is the set of vectors in $\Delta^{\mathcal{S} \times \mathcal{A} \times \mathcal{S}}$ which satisfy conformability of state transitions under transition probabilities; and marginalization:

$$p_b^\infty(j, u', a, k, u'') = p(k, u'' \mid j, u', a) \sum_{\tilde{k}, \tilde{u}''} p_b^\infty(j, u', a, \tilde{k}, \tilde{u}''), \qquad \forall j, u', a, (k, u'') \tag{16}$$

$$\sum_{j,u'} \sum_a p_b^\infty(j, u', a, k, u'') = \sum_{a'} \sum_{\tilde{k}, \tilde{u}''} p_b^\infty(k, u'', a', \tilde{k}, \tilde{u}''), \qquad \forall (k, u'') \tag{17}$$

Lemma 3.1 of [20] states that the two sets are equivalent: if $p_b^\infty(s, u, a) \in$ SAP, and furthermore under the transformation

$$p_b^\infty(j, u', a, k, u'') = p_b^\infty(j, u', a)p(k, u'' \mid j, u', a),$$

then $p_b^\infty(j, u', a, k, u'') \in$ SASP. Every element of SASP can be generated in this manner from some element of SAP. It is this construction that leads to the "information loss" stated formally in the next result which characterizes the marginalized versions of Eqs. (15) to (17) in terms of the marginalized weight of the reparametrization, $g_k(a \mid j)$.

**Lemma 3.** *The marginalized version of Eq. (15) is*

$$p_b^\infty(k) = \sum_a \sum_j p_b^\infty(j)p(k \mid j, a)g_k(a \mid j)^{-1}, \ \forall k \in \mathcal{S} \tag{18}$$

*The marginalized version of Eq. (16) enforces conformability of $g_k(a \mid j)$ for the true non-identifiable transitions,*

$$p(k \mid j, a) = p_b^\infty(a, k \mid j)g_k(a \mid j), \ \forall j, a, k \tag{19}$$

*and of Eq. (17) is the conditional control variate,*

$$p_b^\infty(k \mid a) = \sum_j p_b^\infty(j, a, k)g_k(a \mid j), \ \forall k, a \tag{20}$$

*Proof of Lemma 3.* **Marginalizing Eq. (16):**

Starting from the compatibility restriction with the observed empirical state-action frequencies: $\sum_u p_b^\infty(j, u)\pi_b(a \mid j, u)p(k, u' \mid j, u, a) = p_b^\infty(j, a, k, u')$ and marginalizing over $u'$:

$$\sum_u p_b^\infty(j, u)\pi_b(a \mid j, u)p(k \mid j, u, a) = \sum_u p_b^\infty(j, a, k)$$

$$p(k \mid j, a)p_b^\infty(j) \sum_u p_b^\infty(u \mid j)\pi_b(a \mid j, u)\frac{p(k \mid j, u, a)}{p(k \mid j, a)} = p_b^\infty(j, a, k)$$

$$p(k \mid j, a)g_k(a \mid j)^{-1} = p_b^\infty(a, k \mid j)$$

where $p(k \mid j, a) = \sum_u p(k \mid j, u, a)p_b^\infty(u \mid j)$ and $p_b^\infty(a, k \mid j) = \frac{p_b^\infty(j, a, k)}{p_b^\infty(j)}$.

This leads to the conformability requirement of $g_k(a \mid j)$ for the marginal transition probabilities with respect to the observational joint distribution.

$$p(k \mid j, a) = p_b^\infty(a, k \mid j)g_k(a \mid j) \tag{21}$$

which can also be derived as a marginalization of Eq. (16):

$$\sum_{u,u'} p_b^\infty(j, u, a, k, u') = \sum_{u,u'} p(k, u' \mid j, u, a) \sum_{\tilde{k}, \tilde{u}'} p_b^\infty(j, u, a, \tilde{k}, \tilde{u}'), \qquad \forall j, k$$

**Marginalizing Eq. (17):**

Recall that from the forward decomposition of joint distribution with respect to the transition from $j, u \rightarrow k, u'$, but conditioning on current state and action, we have that:

$$p_b^\infty(k, u' \mid a, j, u) = \frac{p_b^\infty(j, u, a, k, u')}{\pi_a(j, u)p_b^\infty(j, u)}$$

Then marginalize the definition of $p_b^\infty(k, u' \mid a)$ over $u'$ to obtain $p_b^\infty(k \mid a)$:

$$\sum_{u'} p_b^\infty(k, u' \mid a) = \sum_{u'} \sum_{j,u} p_b^\infty(k, u' \mid a, j, u)p_b^\infty(j, u) = \sum_{u'} \sum_{j,u} \frac{p_b^\infty(j, u, a, k, u')p_b^\infty(j, u)}{\pi(a \mid j, u)p_b^\infty(j, u)}$$

$$= \sum_j p_b^\infty(j, a, k) \sum_u \frac{p_b^\infty(j, u, a \mid k)}{p_b^\infty(j, a \mid k)\pi(a \mid j, u)}$$

$$= \sum_j p_b^\infty(j, a, k)g_k(a \mid j)$$

Therefore:

$$p_b^\infty(k \mid a) = \sum_j p_b^\infty(j, a, k) g_k(a \mid j), \forall k, a$$

□

In order to interpret *which* of these constraints are observable implications and which are ultimately informative, we next leverage a structural characterization that $g_k(a \mid j)$ can be interpreted as the function which renders the transition probabilities conformable to the joint distribution. Its proof is of independent interest in establishing the relationship to robust MDPs. We introduce the *biased* marginalized transition probabilities $\tilde{p}(s' \mid s, a)$, which would be obtained from naive estimation from the observational joint distribution:

$$\tilde{p}(s' \mid s, a) := \frac{p_b^\infty(s, a, s')}{p_b^\infty(s)\, \mathbb{E}[\pi_b(a \mid s, u) \mid s]} = p_b^\infty(s)^{-1} \sum_u \frac{p_b^\infty(s, u, a, s')}{\mathbb{E}[\pi_b(a \mid s, u) \mid s]}$$

These are biased estimates because they do not appropriately account for the transitions under the true $\pi_b(s, u)$ policy, only its marginalization over $s$.

**Lemma 4.**

$$\tilde{p}(k \mid j, a)\pi_b(a \mid j) = p(k \mid j, a)g_k(a \mid j)$$

*Proof of Lemma 4.* With full information, the transition probabilities could be estimated as

$$p(s', u' \mid s, u, a) = \frac{p_b^\infty(s, u, a, s', u')}{\pi_b(a \mid s, u) p_b^\infty(s, u)}$$

and similarly, the marginalized transition probabilities as $\frac{p_b^\infty(s, u, a, s')}{\pi_b(a \mid s, u) p_b^\infty(s, u)} = p(s' \mid s, u, a)$.

A model-based perspective would partially identify the transition matrix under $\pi_e$, deduce the bounds of $p(s' \mid s, a)$ relative to $\tilde{p}(s' \mid s, a)$. Note that under Assumption 2,

$$p(s' \mid s, a) = \sum_u p(s' \mid s, u, a) p_b^\infty(u) = \sum_u \frac{p_b^\infty(s, u, a, s') p_b^\infty(u)}{\pi_b(a \mid s, u) p_b^\infty(s, u)} = p_b^\infty(s)^{-1} \sum_u \frac{p_b^\infty(s, u, a, s')}{\pi_b(a \mid s, u)}$$

(22)

and further the distribution on unobserved confounders is independent of the policy, $p_b^\infty(u) = p_e^\infty(u)$.

$$\tilde{p}(s' \mid s, a) = \frac{p_b^\infty(s, a, s')}{p_b^\infty(s)\, \mathbb{E}[\pi_b(a \mid s, u) \mid s]} = p_b^\infty(s)^{-1} \sum_u \frac{p_b^\infty(s, u, a, s')}{\mathbb{E}[\pi_b(a \mid s, u) \mid s]} \qquad (23)$$

Combining Equations (22) and (23) yields the statement of the lemma.

□

Lemma 4 shows that the constraints in Equations (18) and (19) are uninformative: further restricting $\tilde{p}(s' \mid a, s)$ within the given range of $p(s' \mid a, s)$ is redundant. Another interpretation is that $g_k(a \mid j)$ are precisely the weights which render the observed stationary occupancy distribution $p_b^\infty(s, a, s')$ conformable under the *unobserved* true marginal transition probabilities.

Proposition 1 follows from Lemmas 3 and 4.

□

*Proof of Proposition 2.* We verify that optimizing over $F(w) = 0 \iff w \in \Theta, \mathbb{E}[w] = 1$ by first showing the reparametrization of $F(w)$ with respect to $g$, and then leveraging the characterization of Proposition 1 and sharpness argument to verify that $F(w) = 0 \iff w \in \Theta, \mathbb{E}[w] = 1$.

**Step 1: Proving the reparametrization of $F(w)$ with respect to $g_k(a \mid j)$, and reformulating $g_k(a \mid j)$.**

We first expand the sample expectations for the estimating equation of Lemma 2:

$$\sum_j \sum_{i=1}^{N} \sum_{t=0}^{T} \sum_{a,u} \frac{\mathbb{I}[(s_t^{(i)} = j, u_t^{(i)} = u), a_t^{(i)} = a, s_{t+1}^{(i)} = k]}{p(s_{t+1}^{(i)} = k)} \left( w(k) - w(j)\beta^{(i)}(a \mid j, u)\pi_e(a \mid j) \right) = 0, \forall k$$

Rewrite as an expectation with respect to the observational (identifiable) joint distribution $p_b^\infty(j, a \mid k)$, taking limits as $T \to \infty$, $N \to \infty$ and multiplying by $\frac{p_b^\infty(j,a|k)}{p_b^\infty(j,a|k)} = 1$:

$$w(k) - \sum_j w(j) \sum_a \pi_e(a \mid j) p_b^\infty(j, a \mid k) \underbrace{\sum_u \frac{p_b^\infty(j, u, a \mid k)}{p_b^\infty(j, a \mid k)} \beta(a \mid j, u)}_{g_k(j,a;W)} = 0, \forall k$$

Therefore, dependence on $\beta$ arises only through the marginalized weight $g_k(a \mid j)$:

$$g_{s'}(a \mid s) = \sum_u \frac{p_b^{(\infty)}(j, u, a \mid k)}{p_b^\infty(j, a \mid k)} \beta(a \mid j, u)$$

However, the marginalized weight depends on the unknown data-generating joint distribution $p_b^{(\infty)}(j, u, a \mid k)$; and so it is unclear how to optimize over it. Next, we show that *optimizing over* $g_k(a \mid j)$, which are the unknown inverse weights $\beta(a \mid j, u)$ convolved with an unknown density, is almost *equivalent* to optimizing over the set of weights $\mathcal{B}$, up to a moment constraint on ensuring that the implied full-information transition probabilities are valid probability distributions, e.g. that $\sum_k p(k \mid j, u, a) = 1$. We first we show that it is equivalent to optimize over $g_k(a \mid j)$ over the same bounds, though this may not enforce the restriction $\sum_k p(k \mid j, u, a) = 1$. In the next step, we will argue that this restriction to valid transition probabilities is enforced by the feasibility of $g$ for the estimating equation.

Define

$$\tilde{\mathcal{B}}' := \{g \in \mathbb{R}_+^{|\mathcal{S}||\mathcal{A}||\mathcal{S}|} : \exists \beta \in \mathcal{B} \text{ such that } g_{s'}(a \mid s) = \sum_u \frac{p_b^{(\infty)}(j, u, a \mid k)}{p_b^{(\infty)}(j, a \mid k)} \beta\}$$

as the ambiguity region of $g_k(a \mid j)$ induced by restrictions on $\beta \in \mathcal{B}$. We show how to identify elements $\tilde{\beta} \in \tilde{\mathcal{B}}'$ with corresponding elements $\beta \in \mathcal{B}$. Although $p_b^{(\infty)}(j, u, a \mid k)$ is not identifiable from observed data, its marginalization over $u$, $p_b^{(\infty)}(j, a \mid k)$, is identifiable, so we can partially identify $\tilde{\mathcal{B}}'$ as follows:

$$\tilde{\mathcal{B}}' = \left\{ g \in \mathbb{R}_+^{|\mathcal{S}||\mathcal{A}||\mathcal{S}|} : \begin{array}{c} g_{s'}(a \mid s) = \sum_u \frac{p_b^{(\infty)}(j,u,a|k)}{p_b^{(\infty)}(j,a|k)} \beta(a \mid j, u) \\ \sum_u p_b^{(\infty)}(j, u, a \mid k) = p_b^{(\infty)}(j, a \mid k) \\ 0 \le p_b^{(\infty)}(j, u, a \mid k) \le 1 \\ \beta \in \mathcal{B} \end{array} \right\}$$

A simple reparametrization with respect to $q_{j,u,a|k} := \frac{p_b^{(\infty)}(j,u,a|k)}{p_b^{(\infty)}(j,a|k)}$ shows that optimizing over $\tilde{\mathcal{B}}'$ is equivalent to optimizing over elements of $\mathcal{B}$ averaged by unknown weights on the simplex. In the following, suppress dependence of $q_{j,u,a|k}\beta(a \mid j, u)$ on $a, j$ for brevity, and let $q_{j,\cdot,a,|k}$ denote the vector .

$$\tilde{\mathcal{B}}' = \left\{ q_{j,\cdot,a,|k}^\top \beta \ : \ \beta \in \mathcal{B}; \ \ q_{j,\cdot,a,|k}^\top \mathbf{1} = 1, \ 0 \le q_{j,\cdot,a,|k} \le 1, \ \forall j, a, k \right\}$$

In particular this suggests that $g_k'(a \mid j) \in \mathcal{B}$ since by convexity of $\mathcal{B}$, we can map $g_k'(a \mid j) \in \tilde{\mathcal{B}}'$ to some $\beta \in \mathcal{B}$; $q$ are the convex combination weights. In the other direction, clearly any $\beta' \in \mathcal{B}$ is realizable by a $q$ which is a Dirac measure which selects $\beta'$, so $\beta' \in \tilde{\mathcal{B}}'$.

Lastly, we directly verify the control variate property (Eq. (7) corresponding to Eq. (3)) that $\sum_j \sum_k p_b^{(\infty)}(j, a \mid k) p_b(k) g_k(a \mid j) = 1$:

$$
\begin{aligned}
\sum_j p_b^{(\infty)}(j, a, k) g_k(j, a) &= \sum_{j,u} p_b^{(\infty)}(j, a \mid k) p_b^{\infty}(k) \frac{p_b^{(\infty)}(j, u, a \mid k)}{p_b^{(\infty)}(j, a \mid k)} \beta(a \mid j, u) \\
&= \sum_{j,u} \frac{p_b^{(\infty)}(j, u) \pi_b(a \mid j, u) p(k \mid j, u, a)}{p_b^{\infty}(k)} p_b^{\infty}(k) \beta(a \mid j, u) \\
&= \sum_{j,u} p_b^{\infty}(k, j, u \mid a)
\end{aligned}
$$

so that we verify the action-marginal control variate, $\sum_k \sum_j p_b^{(\infty)}(j, a, k) g_k(a \mid j) = 1$. Note that this further implies Eq. (7):

$$
\sum_j p_b^{(\infty)}(j, a, k) g_k(j, a) = \sum_{j,u} p_b^{\infty}(k, j, u \mid a) = p_b^{\infty}(k \mid a)
$$

Finally, to help interpret $g_k(a \mid j)$, we may further simplify and observe that

$$
\begin{aligned}
g_k(a \mid j) &= \sum_u \frac{p_b^{(\infty)}((j, u), a \mid k)}{p_b^{(\infty)}(j, a \mid k)} \beta(a \mid j, u) = \frac{\sum_u p_b^{\infty}(u \mid j) p(k \mid j, u, a)}{\sum_u p_b^{\infty}(u \mid j) p(k \mid j, u, a) \pi_b(a \mid j, u)} \\
&= \frac{p(k \mid j, a)}{\sum_u p_b^{\infty}(u \mid j) p(k \mid j, u, a) \pi_b(a \mid j, u)} \\
&= \frac{1}{\sum_u \pi_b(a \mid j, u) \frac{p_b^{\infty}(u \mid j) p(k \mid j, u, a)}{p(k \mid j, a)}}
\end{aligned}
$$

Note that in this step, we did *not* require the extremal $g_k(a \mid j)$ to be achieved by valid unobserved transition probabilities such that $\sum_k p(k \mid j, u, a) = 1$. To do so directly would introduce technical difficulties in effectively requiring a bilinear formulation. Instead, we have shown this one-to-one correspondence of $\tilde{\mathcal{B}}'$ with $\beta \in \tilde{\mathcal{B}}$ occurs for the marginalized weights, which are not further required to correspond to valid adversarial transition probability distribution. In the next steps, we argue that feasibility of $g \in \tilde{\mathcal{B}}'$ for $F(w) \leq 0$ ensures this compatibility with valid transition matrices.

**Step 2: Proving $F(w) \leq 0, \mathbb{E}[w] = 1 \implies w \in \Theta$:**

Proposition 1 shows that the specification of $\tilde{\mathcal{B}}$ exhausts the observable implications of the sharp full information polytope of all limiting state-action-state occupancy probabilities. It remains to show that $w$ is feasible for some $g_{s'}(a \mid s) \in \mathcal{B}$ iff $g_k(a \mid j)$ satisfies Eq. (19), $p(k \mid j, a) = p_b^{\infty}(a, k \mid j) g_k(a \mid j)$, $\forall j, a, k$.

First we show that $F(w) \leq 0, \mathbb{E}[w] = 1$ implies $w \in \Theta$. Suppose $g$ is feasible for the estimating equation:

$$
w(k) - \sum_j w(j) \sum_a \pi_e(a \mid j) p_b^{\infty}(j, a \mid k) g_k(a \mid j) = 0, \forall k \tag{24}
$$

Next we verify that $g_k(a \mid j)$ satisfying Eq. (19) is feasible for the estimating equation for $w$, Eq. (24). Note that $g_k a \mid j$ corresponding to underlying transitions which do not satisfy $\sum_k p(k \mid j, u, a) = 1$ cannot satisfy Eq. (19). By Bayes' rule and conformability of $g_k(a \mid j)$ for $p(k \mid j, a)$,

$$
p_b^{\infty}(j, a \mid k) = \frac{p(k \mid j, a) p_b^{\infty}(j) g_k(a \mid j)^{-1}}{p_b^{\infty}(k)}, \tag{25}
$$

so that we can verify the estimating equation holds with respect to the true marginalized transition dynamics under $\pi_e$:

$$
w(k) p_b^{\infty}(k) - \sum_j w(j) p_b^{\infty}(j) \sum_a \pi_e(a \mid j) p(k \mid j, a) = 0, \forall k
$$

Markovianness of the induced MDP under the *true* marginal transition probabilities $p(k \mid j, a)$ (follows since $p(k \mid j, a)$ corresponds to the $p^\infty(u)$-occupancy-weighted aggregation to $\mathcal{S}$ under Assumption 2), then $w(k)p_b^\infty(k)$ is proportional to the invariant measure on $\mathcal{S}$.

**Step 3: Proving $w \in \Theta \implies F(w) = 0, \mathbb{E}[w] = 1$:**

Next we show the other direction, that $w$ feasible for Equation (24) for some $g_k(a \mid j)$ implies that $g_k(a \mid j)$ satisfies Equation (19). This direction follows once we identify $w(s)p_b^\infty(s)$ feasible for Equation (24) uniquely with $p_e^\infty(s)$, which follows from Assumption 1. By Equation (25), feasibility implies that $w(s)p_b^\infty(s)$ satisfies compatibilty under $\pi_e$ for $g_k(a \mid j)$. Uniqueness of the density ratio implies that compatibility must also hold under $\pi_b$. $\qquad\square$

*Proof of Theorem 1.* Theorem 1 is a consequence of Proposition 1 and that computing the support function of a set (e.g. optimizing an arbitrary linear objective over $\Theta$) is equivalent to optimizing over the convex hull of $\Theta$ [32], $\text{conv}(\Theta)$. Convexity of the interval and of $\text{conv}(\Theta)$ yields sharpness. $\quad\square$

**Relationship between Lemma 4 and robust MDPs**

**Remark 4.** We can define an ambiguity set for marginal transition probabilities on $\mathcal{S}$ for $s, a$ state-action pairs, $\mathcal{P}_{s' \mid s, a}$:

$$P(\cdot \mid s, a) \in \mathcal{P}_{s' \mid s, a} := \left\{ P_{sa} \in \Delta^{|\mathcal{S}|} : \exists \beta \in \mathcal{B} \text{ such that } P_{sa}(k) = \sum_u \beta \cdot \frac{p_b^\infty(s, u, a, k)}{p_b^\infty(s)}, \forall k \in \mathcal{S} \right\}$$

By an analogous construction as in Proposition 2, without additional restrictions on the variation of the unobserved joint visitation distribution $p_b^\infty(s, u, a, s')$,

$$\mathcal{P}_{s' \mid s, a} := \left\{ P_{sa} \in \Delta^{|\mathcal{S}|} : \exists g \in \tilde{\mathcal{B}} \text{ s.t. } P(s' \mid s, a) = g_{s'}(a \mid s) \cdot \tilde{P}(s' \mid s, a) \right\}$$

However, due to the restrictions on $g_k(a \mid j)$ corresponding to valid probability distributions, as well as the restrictions that $P(s' \mid s, a)$ corresponds to valid probability distributions, the feasible implicit ambiguity set on transition sets is not merely the union of $\mathcal{P}_{s' \mid s, a}$ for all $s, a$. Instead, the valid ambiguity set combines the restrictions induced by the bounds assumptions of $\beta$ over $P_{sa} \in [\Delta^{|\mathcal{S}|}]^{|\mathcal{S}| \times |\mathcal{A}|}$, with $P_{s' \mid s, a} = p(s' \mid s, a)$, and the observable implications of Proposition 1.

$$\mathcal{P}_{\text{feas}} = \left\{ P \in [\Delta^{|\mathcal{S}|}]^{|\mathcal{S}| \times |\mathcal{A}|} : \begin{array}{ll} \exists\, g \in \mathcal{B}, w \text{ s.t.} & \\ P(k \mid j, a) = g_{s'}(a \mid s) \cdot \pi(a \mid j)\tilde{p}(k \mid j, a), & \forall j, a, k \\ p_b^\infty(k \mid a) = \sum_j p_b^\infty(j, a, k) g_k(a \mid j), & \forall k, a \\ \mathbb{E}[\pi_e(a \mid s)w(s)g_k(a \mid s) \mid s' = k] = w(k) \;\; \forall k, & \\ \mathbb{E}[w(s)] = 1 & \end{array} \right\} \quad (26)$$

Notably, the inverse probability restrictions on $g_k(a \mid j)$ render such an ambiguity set *non-rectangular* over states and actions, while the requirement of compatibility (enforced by the estimating equations) additionally introduces nonconvexity.

## C  Proofs: Optimization and algorithms

*Proof of Proposition 3.* A feasibility oracle for $\Theta$, for a given $w$, is given by checking the *existence* of $g \in \mathcal{W}$ satisfying the moment condition. For brevity, let $m_k(w, g)$ denote the $k$th moment restriction:

$$m_k(w, g) := \sum_{j, a} p_b^{(\infty)}(j, a \mid k) w(j) \pi_e(a \mid j) g_{s'}(a \mid s) - w(k)$$

$$F(w) := \min_{g \in \tilde{\mathcal{B}}} \sum_k |m_j(w, g)|$$

We will generate a disjunctive reformulation of $F(w)$ by appealing to a different lifting of the $\ell_1$ norm that enumerates the possible sign patterns on $\{-1, 1\}^{|\mathcal{S}|}$; the next lemma briefly verifies equivalence.

**Lemma 5.**

$$\min\{\sum_j \lambda_j m_k(w,g) \colon g \in \tilde{\mathcal{B}}\} = 0, \forall \lambda_i \in \{-1,1\}^p \iff \min\{\sum_j |m_k(w,g)| \colon g \in \tilde{\mathcal{B}}\} = 0$$

We next briefly introduce the disjunctive programming framework.

**Preliminaries for disjunctive programs**   First we introduce disjunctive programs with generic notation [3]. A disjunctive program optimizes over the union of polyhedra. A disjunctive program is of the form: $\min\{cx | Ax \geqslant a_0, x \geqslant 0, x \in L\}$ where the logical conditions $x \in L$ can be represented by the disjunctive normal form or the conjunctive normal form,

$$\{Ax \geqslant a_0, \quad x \geqslant 0; \underset{i \in Q_j}{\vee} \left(d^i x \geqslant d_{i0}\right), \quad j \in S\},$$

or equivalently, to make the conjunctions apparent,

$$\begin{bmatrix} Ax \geq a_0 \\ x \geq 0 \end{bmatrix} \wedge [\underset{i \in Q_0}{\vee} \left(d^i x \geqslant d_{i0}\right)] \wedge \cdots \wedge [\underset{i \in Q_{|S|}}{\vee} \left(d^i x \geqslant d_{i0}\right)]$$

The linear programming equivalent of a disjunctive normal form is given by Theorem 2.1 of [3]. It generically provides the linear programming formulation of a disjunctive form, $F = \left\{x \in R^n | \bigvee_{h \in Q} \left(A^h x \geqslant a_0^h, x \geqslant 0\right)\right\}$, as

$$\text{clconv } F = \left\{x \in R^n \middle| \begin{array}{ll} x = \sum_{h \in Q^*} \xi^h, & \\ A^h \xi^h - a_0^h \xi_0^h \geqslant 0, & h \in Q^* \\ \sum_{h \in Q^*} \xi_0^h = 1, & \left(\xi^h, \tilde{\xi}_0^h\right) \geqslant 0, h \in Q^* \end{array}\right\},$$

where $Q^*$ is the restriction of $Q$ to nonempty disjunctions.

**Reformulating as a disjunctive program.**   Using Lemma 5, we can rewrite (10), where $X(\mathcal{B})$ denotes the set of extreme points of polytope $\mathcal{B}$:

$$\max\left\{\mathbb{E}[w(s)\Phi(s)] \colon \quad \forall \lambda \in \{-1,1\}^{|\mathcal{S}|}, \ \exists \ g \in X(\tilde{\mathcal{B}}) \text{ s.t. } \sum_k \lambda_k m_k(w,g) = 0\right\}$$

and therefore in the disjunctive syntax,

$$\max\left\{\mathbb{E}[w(s)\Phi(s)] \colon \underset{\lambda \in \{-1,1\}^{|\mathcal{S}|}}{\wedge} \left(\underset{g \in X(\tilde{\mathcal{B}})}{\vee} \text{ s.t. } \lambda_k m_k(w,g) = 0\right)\right\}$$

Next, by applying the distributive property of conjunctions and disjunctions ([3]), we can express the conjunctive form into the disjunctive form, and then apply Theorem 2.1 to obtain the corresponding linear programming representation. Note that this operation generates exponentially many unions of disjunctions (each also exponential in cardinality of state space), and therefore admits a overall superexponentially-sized program. □

*Proof of Lemma 5.* For brevity, denote

$$P(\lambda) := \min\{\sum_j \lambda_j m_k(w,g) \colon g \in \tilde{\mathcal{B}}\}$$

$$P(\ell_1) := \min\{\sum_j |m_k(w,g)| \colon g \in \tilde{\mathcal{B}}\}$$

so that the lemma can be stated as:

$$P(\lambda) = 0, \forall \lambda \in \{-1,1\}^p \iff P(\ell_1) = 0$$

First we argue $\impliedby$ : Suppose not, where $\{\min \sum_j |m_k(w,g)| = 0, \forall j\}$ is true (and achieved by some $w^*$) but there is some $\lambda^*$ such that $\min\{\sum_j \lambda_j^* m_k(w,\lambda) \colon a \leq w \leq b\} > 0$. But $g^*$ was

feasible for $P(\lambda^*)$, and $P(\ell_1) \leq 0$ implies that $m_k(w, g^*) = 0, \forall j$, so we could further reduce the value of $P(\lambda^*)$ at feasible $g^*$, contradicting optimality of $\lambda^*$ at a strictly positive value.

In the other direction, $\implies$, suppose by way of contradiction that $P(\lambda) \leq 0, \quad \forall \lambda$ but $P(\ell_1) > 0$ (equivalently, that $g$ is infeasible for $\mathcal{B}$). Let $\lambda^*_{\ell_1}$ be the particular sign pattern which achieves strict nonnegativity at optimality for $P(\ell_1)$. Then, the optimal value of $P(\lambda)$ at this particular choice of $\lambda^*_{\ell_1}$ is also strictly positive, $P(\lambda^*_{\ell_1}) = P(\ell_1) > 0$ since it induces the same optimization objective over the same feasible set of weights, $\mathcal{B}$.

By uniqueness of $w$ for feasible values of the unknown weights $g_k$ (by properties of ergodicity and uniqueness of the stationary distribution), it is not a relaxation to optimize over different weights for each $\lambda$ value. $\qquad\square$

*Proof of Proposition 4.* We recall the notation which encodes the estimating equation as the matrix $A$: we introduce the instrument functions $\phi_s, \phi_{s'} \in \mathbb{R}^{|\mathcal{S}|\times 1}$, random (row) vectors which are one-hot indicators for the state random variable $s, s'$ taking on each value, $\phi_s = [\mathbb{I}[s=0] \quad \ldots \quad \mathbb{I}[s=|\mathcal{S}|]]$. Let $A(g) = \mathbb{E}[\phi_{s'}(\pi_e(a \mid s)g_{s'}(a \mid s)\phi_s - \phi_{s'})^\top]$ and $b_s = p^\infty_b(s)$. The set of $g \in \tilde{\mathcal{B}}$ that admit a feasible solution to the estimating equation for some $w \in \Theta$ is $\psi := \{g \in \tilde{\mathcal{B}} : \exists \ w \geq 0 \text{ s.t. } A(g)w = 0, b^\top w = 1\}$

$A$ has rank $|\mathcal{S}| - 1$ *if* $g$ is feasible since satisfying the conformability constraint Equations (16) and (20) implies linear dependence on the rows of $A$:

$$\sum_k \left(w_k - \sum_{j,a} p^\infty_b(a, j \mid k)\pi_e(a \mid j)g_k(a \mid j)w_j\right) = 0$$

Define $\tilde{A}(g)$ by replacing the last row of $A(g)$ by $b$ and let $v = (0, \ldots, 0, 1)$. Then $w = \tilde{A}^{-1}v$ which results in the following program, with $v = \begin{bmatrix} 0_{|\mathcal{S}|-1} & 1 \end{bmatrix}^\top$:

$$\inf \ / \ \sup \left\{\varphi^\top \tilde{A}^{-1}v \ : \ g \in \psi\right\}$$

Partial derivatives of the matrix-valued function of $g_k(a \mid j)$ follow from the matrix chain rule, where $J_{j,k}$ is a one-hot matrix with a 1 in the $j, k$ entry and 0 everywhere else:

$$\begin{aligned}
\frac{\partial \varphi^\top \tilde{A}^{-1}v}{\partial g_k(a \mid j)} &= \sum_{i,j} \frac{\partial \varphi^\top \tilde{A}^{-1}v}{\partial \tilde{A}_{i,j}} \frac{\partial \tilde{A}_{i,j}}{\partial g_k(a \mid j)} \\
&= -\sum_{i,j} \tilde{A}^{-\top}_{i,j}\varphi v^\top \tilde{A}^{-\top}_{i,j}(\mathbb{I}[j \neq |\mathcal{S}|]\pi^e_{a,j} \ \overset{\infty_b}{p}_{j,a,k} \ J_{j,k}) \\
&= -(\mathbb{I}[j \neq |\mathcal{S}|]\pi^e_{a,j} \ \overset{\infty_b}{p}_{j,a,k})(\tilde{A}^{-\top}\varphi w^\top)_{i,j}
\end{aligned}$$

$\qquad\square$

# D   Proof of Theorem 3 (statistical consistency)

Consistency follows from stability of the optimization problem in terms of the deviations of empirical probabilities from their population values. The former is a result from variational analysis/stability analysis of linear programs, which is non-standard because the perturbations occur in the constraint matrix *coefficients*. The latter is simply the convergence of empirical probabilities.

We prove statistical consistency by considering a support function estimate that discretizes the space (restricting attention to feasible $w$). We apply a stability analysis argument to argue consistency for every fixed $w$ in the discretization. Since $w$ is in a compact set (the $\mathcal{S}$ simplex), a covering argument over the solution space provides a bound via a union bound over elements of the discretization. Lastly, we bound the approximation error arising from the discretization. While the discretization approach provides a statistical consistency result (in the limit as $n \to \infty$ we assume the discretization grows finer), it is a tool of the analysis but not the algorithmic proposal.

**Preliminaries: Stability Analysis** Consistency follows from stability of the optimization problem in terms of the deviations of empirical probabilities from their population values. The former is a result from variational analysis/stability analysis of linear programs, which is non-standard because the perturbations occur in the constraint matrix *coefficients*. The latter is simply the convergence of empirical probabilities.

Stability analysis establishes convergence in Hausdorff distance between $\hat{\Theta}$, $\Theta$, the partial identification set obtained from optimizing the sample estimating equation vs. from optimizing the population estimating equation. The Hausdorff distance between two sets $A, B \subset \mathbb{R}^d$ is $d_H(A, B) = \max\{\sup_{a \in A} d(a, B), \sup_{b \in B} d(b, A)\}$ where $d(a, B) = \inf_{b \in B} \|a - b\|$; e.g. it measures the furthest distance from an arbitrary point in one of the sets to its closest neighbor in the other set.

The main stability analysis result we use is Theorem 1 of [31]. To help keep the presentation of the theorem self-contained, we state some preliminary notation. The paper considers the general case of a system of linear inequalities, where $A$ is a continuous linear operator from $X$ into $Y$ which are real Banach spaces, and $K$ is a nonempty closed convex cone in $Y$. We study

$$Ax \leq_K b, \forall x \in C \tag{27}$$

with $C \subseteq X$ a convenience set to represent *unperturbed* constraints. We want to ascertain the *stability region* of the solution set $G$, which implies that for *each* $x_0 \in G$, for some positive number $\beta$, and for any continuous linear operator $A'\colon X \mapsto Y$ and any $b' \in Y$, the distance from $x_0$ to the solution set of the perturbed system,

$$A'x \leq_K b', \tag{28}$$

is bounded by $\beta\rho(x_0)$, with $\rho(x)$ being the residual vector,

$$\rho(x) := d(b' - A'x, K) := \inf\{\|b' - A'x - k\| \mid k \in K\}.$$

For a more concise statement of the main stability analysis result, we introduce the augmented operator with an auxiliary dimension to homogenize the system $Q\colon X \times \mathbb{R} \mapsto Y$: for finite-dimensional systems of linear equations, this is the usual homogenization.

$$Q(\begin{bmatrix} x \\ \xi \end{bmatrix}) = \begin{cases} \begin{bmatrix} A & -b \end{bmatrix} \begin{bmatrix} x \\ \xi \end{bmatrix} + K & , \begin{bmatrix} x & \xi \end{bmatrix}^\top \in P, \\ \infty & \begin{bmatrix} x & \xi \end{bmatrix}^\top, \notin P \end{cases}$$

Now, under this notation, $x \in C$ satisfies $Ax \leq_K b, \forall x \in C$ iff $0 \in Q(\begin{bmatrix} x \\ \xi \end{bmatrix})$. The result will leverage properties of linear operators as special cases of convex processes (which are themselves multivalued functions between two linear spaces): If $T$ carries $X$ into $Y$, with $X, Y$ normed linear spaces, then the *inverse* of $T$ is $T^{-1}$, which is defined for $y \in \mathcal{Y}$ by

$$T^{-1}y := \{x \mid y \in Tx\}.$$

$T$ is closed if $\mathrm{gph}(T) := \{(x, y) \mid y \in Tx\}$ is closed on product space $X \times Y$. The norm of $T$ is operator norm.

This approach then allows us to identify another linear operator which parametrizes the *perturbation* $\Delta(\begin{bmatrix} x \\ \xi \end{bmatrix})$ defined analogously to $Q$ but with $(A' - A)x - (b' - b)\xi + K$. The size of the perturbation is measured by the operator norm of this system, and a crude bound is $\|\Delta\| \leq \|A' - A\| + \|b' - b\|$; we also have that $\rho(x) \leq \|\Delta\| \max\{1, \|x\|\}$.

**Assumption 3.** Regularity: $b \in \mathrm{int}\{A(C) + K\}$ and singular otherwise.

The required regularity assumption is similar to strict consistency of [32], which states that $0 \in \mathrm{int}(\mathrm{dom})(G)$, e.g. there exists $u, v$ such that all inequality constraints $f < 0$ hold strictly.

Finally, having introduced the homogenized system $Q$ and the perturbation $\Delta$, we state the required theorem: $Q' = Q + \Delta$ is the perturbed augmented system.

**Theorem 5** (Linear system stability (Theorem 1, [31])). *Suppose that the system Eq. (27) is regular. Then $Q$ is surjective, $\left\|Q^{-1}\right\| < +\infty$, and if $\left\|Q^{-1}\right\|\|\Delta\| < 1$, then is also regular (hence solvable), with $\left\|Q'^{-1}\right\| \leqq \left\|Q^{-1}\right\| / \left(1 - \left\|Q^{-1}\right\|\|\Delta\|\right)$. Further, if $G'$ denotes the solution set of Eq. (28), then for any $x \in C$ with $\left\|Q'^{-1}\right\|\rho(x) < 1$ we have*

$$d\left(x, G'\right) \leqq \left[\frac{\left\|Q'^{-1}\right\|\rho(x)}{1 - \left\|Q'^{-1}\right\|\rho(x)}\right](1 + \|x\|) \tag{29}$$

We now use these results to prove consistency.

*Proof.* Proof of Theorem 3

We consider the proposed support function estimator which conducts a grid search over an $\epsilon$-covering, $\mathcal{E}_w$ covering $\mathbb{R}_+^{|\mathcal{S}|}$ such that $\mathcal{E}_w$ is the smallest set satisfying that $\min_{w'} \colon \|w - w'\|_1 \leq \epsilon \; \forall w$ s.t. $w^\top \vec{1}$. For simplicity, we consider the self-normalized version of the estimator that constrains $w^\top \vec{1} = 1$: convergence for the stochastic constraint $\mathbb{E}[w] = 1$ holds by additionally conditioning on the event $\|p_b^\infty(s)\|_1 \leq \epsilon$ under fast geometric convergence of the stationary distribution.

The discretization-based support function estimator is:

$$\hat{\bar{R}}_e = \max\left\{\mathbb{E}_n[w(s)\Phi(s)] \; : w \in \mathcal{E}_w, \; w \in \hat{\Theta}, w^\top \vec{1} = 1\right\}$$

$$\text{where } \hat{\Theta} := \{w \in \mathcal{E}_w \colon \sum_{j,a} \hat{h}_{j,a,k}(w)g_{s'}(a \mid s) - w(k) = 0, \; \forall k \in \mathcal{S}; g \in \tilde{\mathcal{B}}\}$$

$$\hat{h}_{j,a,k}(w) := \hat{p}_b^{(\infty)}(j, a \mid k)w(j)\pi_e(a \mid j).$$

We will bound the approximation error

$$\left\|\hat{\bar{R}}_e - \overline{R}_e\right\| \leq \left\|\hat{\bar{R}}_e - \hat{\tilde{R}}_e\right\| + \left\|\hat{\tilde{R}}_e - \tilde{\bar{R}}_e\right\| + \left\|\tilde{\bar{R}}_e - \overline{R}_e\right\|$$

with respect to the intermediary terms,

$$\hat{\tilde{R}}_e = \max\left\{\mathbb{E}_n[w(s)\Phi(s)] \; : w \in \mathcal{E}_w, \; w \in \Theta, w^\top \vec{1} = 1\right\}$$

$$\tilde{\bar{R}}_e = \max\left\{\mathbb{E}[w(s)\Phi(s)] \; : \; w \in \mathcal{E}_w, w \in \Theta, w^\top \vec{1} = 1\right\}$$

$$\overline{R}_e = \max\left\{\mathbb{E}[w(s)\Phi(s)] \; : \; w \in \Theta, w^\top \vec{1} = 1\right\}$$

Bounding $\left\|\hat{\tilde{R}}_e - \hat{\tilde{R}}_e\right\|$ follows by verifying stability of the feasibility problem $\{\sum_{j,a} \hat{h}_{j,a,k}(w)g_{s'}(a \mid s) - w(k) = 0, \; \forall k \in \mathcal{S}\}$ for every such value of $w$, taking a union bound over the covering, and then applying stability of $w$ for a given feasible $g$ to bound the objective values.

$G(w), \hat{G}(w)$ correspondingly denote the solution sets (in the space of $g$) of the feasibility program, for a fixed $w$ vector. We first bound $\left\|\hat{\bar{R}}_e - \hat{\tilde{R}}_e\right\|$ by bounding the distance between the solution set for $G(w), \hat{G}(w)$ for a fixed $w$. We then apply the bound for every $w$ on a covering of the $|\mathcal{S}|$ simplex.

To map the problem quantities to the stability analysis notation, let the set of unperturbed constraints be $C := \{x \in \mathcal{B}\}$. Observe that $\left\|Q'^{-1}\right\| = \max_{\|y\| \leq 1}\{\|x\| : Q'x = y, x \in C\} \leq |\mathcal{S}|^2|\mathcal{A}|\nu$ because the *unperturbed* constraints include bounds constraints on $g$.

Bounding $\|\Delta\|$ proceeds by observing that since the perturbation matrix to the coefficients comprises of terms

$$\sum_{j,a} w(j)\pi_e(a \mid j)(\hat{p}_b^\infty(j, a \mid k') - p_b^\infty(j, a \mid k')),$$

applying a crude bound that $w(j), \pi_e(a \mid j) \leq 1$, it is sufficient to bound the operator norm of perturbations as

$$\|\Delta\|_1 \leq \|\hat{p}_b^\infty(s, a \mid s') - p_b^\infty(s, a \mid s')\|_\infty.$$

Consistency of the empirical state-action probabilities for the population state-action probabilities yields the result, e.g. that $\hat{p}_b^\infty(s, a, s') \to_p p_b^\infty(s, a, s')$, and $\hat{p}_b^\infty(s) \to_p p_b^\infty(s)$. (See [20] for quantitative rates). Consider $n$ large enough so that the condition $\left\|Q'^{-1}\right\| \rho(x) < \frac{1}{2}$ holds. Then,

$$d(\hat{x}, G) \le \left[\frac{\|Q'^{-1}\|\rho(x)}{1 - \|Q'^{-1}\|\rho(x)}\right](1 + \|x\|) \le 2(1 + |\mathcal{S}|^2|\mathcal{A}|\nu)^2 \|\hat{p}_b^\infty(s, a \mid s') - p_b^\infty(s, a \mid s')\|_\infty \tag{30}$$

for any $\hat{x} \in \hat{G}$, with probability 1; and analogously for $d(x, \hat{G})$. The above bound holds for a fixed $w$.

We next bound $\left\|\hat{\bar{R}}_e - \hat{\tilde{R}}_e\right\|$. Taking a union bound over finitely many $w \in \mathcal{E}_w$ so that the bound (30) holds uniformly over $\mathcal{E}_w$, there exists $n_1$ large enough such that

$$\max_{w \in \mathcal{E}_w} d(\hat{G}(w), G(w)) \le \delta. \tag{31}$$

This in turn suggests finite-time identification of the feasible set, $\hat{\Theta}(\mathcal{E}_w) = \Theta(\mathcal{E}_w)$, so that $\hat{\bar{R}}_e = \hat{\tilde{R}}_e$.

We next bound $\left\|\hat{\tilde{R}}_e - \tilde{\bar{R}}_e\right\|$. For every $\epsilon'$, by geometric convergence of the stationary distribution $\hat{p}_b^\infty(s) - p_b^\infty(s)$ and taking a union bound over $w \in \mathcal{E}_w$ to establish uniform convergence, there exists $n_2$ large enough so that:

$$\left\|\hat{\tilde{R}}_e - \tilde{\bar{R}}_e\right\| \le \max_{w \in \Theta}\{\mathbb{E}_n[w(s)\Phi(s)] - \mathbb{E}[w(s)\Phi(s)]\} \le \epsilon'.$$

Lastly, $\left\|\tilde{\bar{R}}_e - \bar{R}_e\right\|$ is bounded by the uniform approximation error which is satisfied by the definition of the covering, and bounded state rewards $\Phi(s)$. Therefore, we have that for some $n \ge \max\{n_1, n_2\}$,

$$\left\|\hat{\bar{R}}_e - R_e\right\| \le \epsilon' + \epsilon$$

Therefore we obtain statistical consistency as we take the discretization width $\epsilon \to 0$ as $n \to \infty$.

$\square$

## D.1 Linear function approximation

Note that since the tabular case is a special case of linear function approximation of the state space, our approach also handles the case where $w = \theta^\top s$ is a linear parameter of the state observation.

We introduce some simplifications. $\Psi_k^{i:t,t+1} = (\phi(s_{t+1}^i)\phi(s_t^i)^\top)_k$ is the $k$th row vector (resp. $\Psi_k^{t,t}$) and

$$\bar{\phi}(s) = \frac{1}{NT}\sum_{i=1}^N \sum_{t=1}^T \phi(s_t^i) = \mathbb{E}_N \mathbb{E}_T[\phi(s)]$$

$$\bar{\Psi}^{t+1,t+1} = \frac{1}{NT}\sum_{i=1}^N \sum_{t=1}^T \phi(s_{t+1}^i)\phi(s_t^i)^\top$$

The corresponding finite-sample feasibility oracle for the case $w = \theta^\top s$, drawing on the least-squares representation of Proposition 4, is:

$$F(\theta) := \min_{g \in \mathcal{B}, z \geq 0} \sum_k z_k$$

$$\text{s.t. } z_k \geq \frac{1}{NT} \sum_{i=1}^N \sum_{t=1}^T \left( \pi_{i,t}^e \Psi_k^{i:t,t+1} \theta \cdot g_{i,t} - \Psi_k^{i:t+1,t+1} \theta \right),, \qquad \forall k \in [d]$$

$$z_k \geq -\frac{1}{NT} \sum_{i=1}^N \sum_{t=1}^T \left( \pi_{i,t}^e \Psi_k^{i:t,t+1} \theta \cdot g_{i,t} - \Psi_k^{i:t+1,t+1} \theta \right), \qquad \forall k \in [d]$$

$$l_{i,t} \leq g_{i,t} \leq m_{i,t}, \quad \forall i \in [N], t \in [T]$$

$$\frac{1}{NT} \sum_{i=1}^N \sum_{t=1}^T \mathbb{I}[A_{it} = a'] g_{i,t} = 1, \forall a'$$

To obtain bounds, using $\mathbb{E}_N \mathbb{E}_T$ as shorthand notation for empirical expectations over trajectories and timesteps within trajectories, one then solves:

$$\inf/\sup \ \{\mathbb{E}_N \mathbb{E}_T[\Phi(s)]^\top \theta \colon F(w) \leq 0, \ \mathbb{E}_N \mathbb{E}_T[\phi(s)]^\top \theta\}. \tag{32}$$

since $\mathbb{E}_N \mathbb{E}_T[w(s)] = \mathbb{E}_N \mathbb{E}_T[\phi(s)^\top \top \theta] = \mathbb{E}_N \mathbb{E}_T[\phi(s)]^\top \theta$.

Finally we remark on additional difficulties in the linear function approximation setting that are not encountered in the tabular setting. The computational burden increases because the reparametrization argument of Proposition 2 crucially relied on the discrete distribution of $S$. In practice, introducing a number of bilinear variables which grows with $n$ performs quite poorly in comparison to the re-parametrization. Furthermore, the approach *requires* well-specification of the linear function class for $w$, since we effectively require realizability of a linear parameter for some set of possible inverse propensity weights. Handling model misspecification via a feasibility relaxation for the approximation error of the function class may therefore tradeoff sharpness; we leave this for future work.

## E   Additional Empirics and Details

**Confounded gridworld**   In Figure 4 we include a depiction of the 3x3 gridworld as described in the main text describing the reward structure in greater detail. (If an action moves an agent into a wall, it simply remains in place).

**SDP relaxation**   One possibility is to consider the standard SDP relaxation for nonconvex quadratic programs [24]. Let $x = \begin{bmatrix} g \\ w \end{bmatrix}$, and $P_k$ be the matrix that generates the quadratic form $\sum_{a,j} w(j) p_b^\infty(s, a, s) g_k(a \mid j) \pi_e(a \mid j)$, e.g. we have that $P_{(k,a,j),|\mathcal{S}|^2|\mathcal{A}|+j}^k = \pi_e(a \mid j) p_b^\infty(s, a, s), \forall a, j$. Then a standard relaxation gives that the solution to the following semidefinite program is a lower bound:

$$\max \left\{ \mathbb{E}[w(s)\Phi(s)] \colon \text{Tr}(XP_k) - w(k) p_b^\infty(k) = 0, \forall k; \ g \in \tilde{\mathcal{B}}, \mathbb{E}_b[w(s)] = 1, \begin{bmatrix} X & x \\ x^\top & 0 \end{bmatrix} \succeq 0 \right\}$$

However, the lifting results in a $(|\mathcal{S}|^2|\mathcal{A}|)$ square matrix; given that under mild assumptions, typical SDPs can be solved in $O(n^2 m^{5/2} \log(\epsilon^{-1}))$ where $n$ is dimension of the vector variable and $m$ is dimension of the matrix variable, we incur an intractable $O(|\mathcal{S}|^9|\mathcal{A}|^{9/2})$ scaling overall.

**Details for confounded random walk**   Note that the stationary distribution under $\pi_b$ is, for $s_1, s_2$:

$$\left( \frac{-1 + (1 - 2\pi_{s_2 u_1})(p_{u_1} + p_{u_2}) + \pi_{s_2 u_1}}{-1 + (\pi_{s_2 u_2} - \pi_{s_1 u_1})(1 - 2p_{u_1} - 2p_{u_2})}, \frac{p_{u_1} + p_{u_2} + \pi_{s_1 u_1}(1 - 2p_{u_1} - 2p_{u_2})}{1 + (\pi_{s_1 u_1} - \pi_{s_2 u_1})(1 - 2p_{u_1} - 2p_{u_2})} \right)$$

Figure 8: Comparison of global optimization and Algorithm 1.

**Algorithm 1 vs. global optimization**    We compare the results from using Algorithm 1 (nonconvex projected gradient descent) vs. solving to full optimality by Gurobi. We find that empirically, storing previous solutions and imposing monotonicity (e.g. for any $\Gamma$, taking the max of all previous returned values and the computed values) helps stabilize the optimization.

**Computational details**    A complete description of the data collection process, including sample size.

- Data: bounds computed based on a trajectory with 40000 steps, and a grid of 25 linearly spaced $\Gamma$ values from $\log(\Gamma) \in [0.1, 1.7]$ (equivalently, $\Gamma \in [1.10, 5.47]$).

- Experiments were run on a Macbook Pro with 16gb RAM.

- Packages: Python (numpy/scipy/pandas), Gurobi Version 9