[Reviews · NeurIPS 2020]

Review 1

Summary and Contributions: The authors propose a method for estimating the partial identification set of the value of a policy in an infinite-time-horizon MDP when the observed policy depends on unobserved confounders, in the special case where the distribution of unobserved confounders does not depend on the previous state. In this setting, the authors propose a bounded sensitivity model similar to bounded odds ratio of treatment models in causal inference. They then characterize the partial identification set of importance weights that would yield policy value estimates that are consistent with the observed data in terms of a linear program. To map this to a partial identification set for the scalar value of the policy, the authors discuss an impractical direct optimization approach, and a more practical non-convex optimization approach. The authors also establish that the estimating equations that they derive are consistent. Finally, the authors demonstrate their method on two simple examples.

Strengths: Partial identification techniques are much-needed in policy evaluation and causal inference contexts to provide more honest uncertainty quantification in settings where assumptions sufficient for identification are implausible. There is a real need in the machine learning community to outline problems where the observed data provide no single “right” answer. This paper is a great contribution to address this need in a setting that could have real practical implications. The derivations in the paper are thorough, and the set of non-trivial steps necessary to make the problem tractable are clearly described.

Weaknesses: Although the paper is generally clearly written, a number of the steps are difficult to parse. A bit more prose description of the mathematical objects, for example the marginal weights g_k(a | j) would go a long way to build intuition in the formulation of the method.

Correctness: They appear to be so, although I did not have time to go through all of the derivations in detail.

Clarity: Generally, yes.

Relation to Prior Work: Yes.

Reproducibility: Yes

Additional Feedback: I really enjoyed this paper, so my comments mostly have to do with making the derivations a bit more readable. The main steps that I got hung up on in reading where the marginalization step, moving from weights beta to weights g, and the step where the matrix A(g) is defined. In both cases, I think some prose description of exactly what the transformation is would be helpful. For the weights g, I think the direct interpretation (the last expression in the line defining g_k(a | j) is more intuitive than the definition in terms of beta. It is not obvious how one moves from one to the other (especially with the inverse migrating out of the summation). It might make sense to define g_k in terms as the weights of the marginalized policy direction first, and then to show the correspondence to beta in a small proposition. I think a slightly extended discussion of how the weights g_k compare to pi_b^{-1} would provide some more insight into how to interpret these weights. It seems clear to me that when transitions do not depend on u, all of the g_k end up being equal to pi_b for each k, so the confounding introduces an interesting non-equivalence here. Understanding how to interpret the dependence on k under confounding would provide some useful insight, I think. One thing that made this section difficult to parse was the use of generic indices for current and next states that differed from other parts of the paper. I think the authors should standardize on (s, s’) or (j, k) (personally, I prefer s, s’). For the Non-convex method, I think it might be useful to describe the set of constraints represented by the matrix A(g), rather than describing its construction so explicitly. Specifically, retating equation (5) in terms of g, describing the set of individual constraints this imposes, and then stating that these can be arranged in a matrix might make this more readable. The actual matrix construction could be left for the appendix. Some nitpicks on the experiment plots. For the consistency plot, having a horizontal line at zero would be useful. For the plots of bounds, it might make sense to focus on one particular set of bounds, and plot the others with a lower alpha value (that is, less opacity) so that it is easier to parse the focal example, while the general trend is visible in the background. Clarifying that the left-hand size of the plot corresponds to the estimate assuming no confounding could also be useful. Finally, a question about identifiability: would it be possible to construct an example in the spirit of, say, the Tennenholtz et al paper where the value of the policy is in identifiable, and to show that the estimated bounds collapse? How do the assumptions from that setting map onto the one that you propose? Would they contradict the sensitivity model directly, or would the consequences be observable after computing the bounds?


Review 2

Summary and Contributions: This paper studies the problem of partial identification of causal effects in settings with an infinite horizon. In particular, the authors consider a canonical Markov decision process model with non-time-varying unobserved confounders. The reward function at each stage of intervention is presumed to be known. However, the transition distribution is affected by an unobserved confounder whose values are drawn independently at each timestep. Let S_t, X_t, Y_t denote, respectively, the observed state, action, reward realized at time t. We denote by do(pi) an intervention following a conditional policy pi: S_t -> X_t. This paper focuses on the average causal effect over the infinite horizon T, i.e., R(pi) = lim_{T -> infty} 1/T E[sum_{t=1}^{T} Y_t|do(pi)]. The authors also assume that the Markov process converges and leads to a stationary distribution P(s_t, x_t, s_{t+1}). The goal is to estimate the unknown causal effect R(pi) for an arbitrary policy pi from the stationary distribution P(s_t, x_t, s_{t+1}).

Strengths: In the general settings, the causal effect R(pi) is not uniquely discernible from the observational data, i.e., it is not identifiable. An alternative solution is the partial identification where the goal is to narrow the parameter space of R(pi) from the observational data. This paper follows the partial identification framework from (Kallus and Zhou, 2018) (for short, KZ18), where they assume the access to a sensitive parameter F that characterizes the strength of unobserved confounding to the data-generating policy that determines observed action X_t. The partial identification problem is thus reducible to a series of linear programs. Unfortunately, the size of the resultant linear programs could be super-exponential. The authors then propose an alternative formulation of non-convex optimization program and discuss practical methods to approximate the bounds. As far as I am aware, this paper is the first work that attempts partial identification of causal effects in settings with an infinite horizon. In many decision-making settings, the unobserved confounding is common, and the agent has to make infinitely many decisions. Therefore, the results presented in this paper could find wide applications in learning policies with suitable performance. While the primary bounding strategy follows (KZ18), non-trivial generalization is required in infinite horizon settings.

Weaknesses: While I understand that one has to constrain the time-dependence of unobserved confounders to obtain reasonable bounds, I still have some questions regarding Assumption 2, summarized as follows. 1. Why does the existence of baseline UCs permit Assumption 2? More specifically, could you please show that one could derive the memoryless UC property from the graphical assumptions in Fig1 (b)? 2. While Lemma 1 allows on to derive Assumption 1, its condition is a bit overly restricted. It seems to suggest that the exogenous variables U_t at each timestep are drawn uniformly at random. This condition is somewhat unrealistic in practice. It also appears that Assumption 2 could hold in more general settings, e.g., Fig1(a). Could the authors explain the rationale behind Lemma 1?

Correctness: I haven't checked the details of the proof. However, the main results seem to be reasonable.

Clarity: This paper is clearly-written and well-organized.

Relation to Prior Work: The references and discussion of the related work are sufficient.

Reproducibility: Yes

Additional Feedback: -- POST REBUTTAL -- I have read the authors’ responses and other reviewers’ comments. Overall, I enjoyed reading this paper but was confused with some technical aspects of the results. In particular, it was not clear to me the relationships between Assumption 2 and the graphical models in Fig 1(a,b). After checking proofs in Appendix, I can now see how Assumption 2 is entailed in causal models of Fig 1(a-b). I think the primary source of confusion is due to Lemma 1, which does not seem to be applicable in either Fig 1(a) or (b). To see this, in Fig 1(a), the condition in Lemma 1 does not hold if u’ \neq \tilde{u}’. While the condition of Lemma 1 could hold in Fig 1(b), the deterministic relationship among Us implies that some P(s’,u’|s,u,a) = 0. Such zero value could lead to questions on the validity of Lemma 1 since its proof involves the ratio of a product of P(s’,u’|s,u,a). Nevertheless, the issue of Lemma 1 is a minor one, and should be easily fixable. For example, the authors could rephrase the conditions in Lemma 1 as variables U_i are mutually independent, which explains Fig 1(a). The authors could then have a separate discussion on the validity of Assumption 2 in Fig 1(b): e.g., once U_0 = U_1 = … U_n are fixed, the environment is reducible to a standard MDP; thus, Assumption 2 is entailed. Overall, I believe that the explanation on Assumption 2 and Lemma 1 could be improved, but it does not seem to affect the soundness of the main results in this paper. Having said that, I am willing to increase my score by 1 point, provided that the authors could resolve the issues of Lemma 1 in the future draft.


Review 3

Summary and Contributions: This paper studies the extent of non-identification in off-policy infinite-horizon RL when certain state space variables are unobserved.

Strengths: Understanding the robustness of infinite horizon MDP methods to missing data is an important topic, given the recent popularity of infinite horizon MDP methods in recent years. This paper has significant technical depth, and appears to be addressing a challenging problem.

Weaknesses: The final proposed approach to bound R_e produces results which appear difficult to interpret for a few reasons. One, a common difficulty in partial identification, is choosing the parameter \Gamma to appropriately reflect reality. While this is a fundamental difficulty, and not unique to this paper, the paper lacks any significant discussion about how one may go about choosing the level \Gamma, or what a reasonable range might be. What side information would an analyst need to know to choose an appropriate value? Two, is the assumptions that are made instead of assuming that all state variables are observed. Both Assumption 1 and 2 appear to make untestable assumptions, as they depend on unobserved variables. The paper provides two examples of when Assumption 2 might hold, but does not discuss the implications of Assumption 1 when certain states are unobserved. In particular, Assumption 1 makes assumptions that jointly depend on the unobserved state and the evaluation policy. Can these be checked in any way? Do these put significant constraints on what the evaluation policy can be? Three, the method partially identifies R_e using a non-convex optimization problem that cannot be simultaneously solved efficiently and exactly. Because the paper suggests to solve the problem using approximate projected gradient descent, there is no guarantee that the resulting solution actually includes the entire partially identified region at termination. Therefore, the output may be an upper bound on the lower bound on R_e, or possibly a real (but loose) lower bound on R_e. This significantly weakens the practicality of using this to draw truly robust conclusions.

Correctness: The mathematical results appear to be correct. The empirical results would benefit from verifying both Assumptions 1 and 2 in each setting.

Clarity: The paper is very dense and makes many large mathematical jumps that are only explained in the appendix. It was difficult to follow the main idea due to the multiple reformulations of the optimization problem.

Relation to Prior Work: The related works section suggests that all previous work on unobservables in RL has introduced assumptions that allow identification. It would be appropriate to clarify that Zhang, J. and Bareinboim, E., 2019 (already cited) and Namkoong, H., et al., 2020 (currently missing, title is "Off-policy Policy Evaluation For Sequential Decisions Under Unobserved Confounding") have developed partial identification results in finite-horizon (it seems) settings, as readers may be interested in these settings as well.

Reproducibility: Yes

Additional Feedback:


Review 4

Summary and Contributions: The paper studies off-policy policy evaluation under unobserved confounding where the policy value can not be point-identified. The authors propose to compute an upper bound and a lower bound on the target policy value by computing the support function of the partially identified set that consists in the set of all valid state-action occupancy ratios that agrees with the sensitivity model they consider. They propose as well a computationally efficient approximation to their method based on nonconvex project gradient descent. Finally, they provide some empirical results on gridworld's domain to illustrate the proposed method.

Strengths: I am not familiar with this area of research so I might be not qualified to judge the contribution of this paper. But, overall the proposed methods sounds correct to me. The author first state their main assumption on the unobserved confounding which is satisfied for stationary or baseline confounders. Then, they introduce the sensitivity model. By marginalization, they reparametrize the sensitivity model in term of the marginalized weights g which seems to decreases the number of constraints as they marginalize over the confounders. Then they characterize the "plausible set" of state-action occupancy ratios in term of a linear program that involves an optimization of g. Finally, the proposed method involves a bilevel optimization problem that they approximately solve with nonconvex project gradient descent.

Weaknesses: I think the main limitation of this work is the that it relies on the discrete nature of S (which might be okay) and the proposed method scales necessarily with |S|^2 |A|, a thing that we cannot afford even for moderately large MDP. I think also that it would be great if the author provide state more clearly the computational complexity of their method.

Correctness: The method looks sounded to me.

Clarity: I am really familiar with the causal inference's terminology and some parts are not very clear for non initiated readers like me. However, the paper looks overall in good shape.

Relation to Prior Work: The connection to related works in both OPE and causal inference literature seems well discussed in introduction as well as in related work section.

Reproducibility: Yes

Additional Feedback:

[Author Response · NeurIPS 2020]

We thank the reviewers for thoughtful reviews and encouraging comments. We respond only to questions and concerns.

(R1) "Although the paper is generally clearly written ...I really enjoyed this paper, so my comments mostly have to do
with making the derivations a bit more readable.": Thanks for the helpful feedback. We will use it to *further* improve
clarity. We will, e.g., include more in-words descriptions of definitions and results and be consistent on $(s, s')$ vs $(j, k)$.

(R1) "Finally, a question about identifiability ...": Great question. If we impose additional assumptions on $M, \pi_b$ in
the definition of $\Theta$ in Sec 3.3 then the set shrinks. Tennenholtz et al. study a particular set of assumptions (that we
discuss on page 8) that would make the set shrink to a *point*. The assumptions imply in a certain sense a view on every
confounder; we work without these assumptions where policy value is *not* point-identifiable. Note that bounds (i.e., $\Theta$)
would only collapse if you impose the assumptions *a priori* on $M, \pi_b$ – no method can automatically detect the validity
of identifying assumptions as they must be imposed on the distribution of *unobserved* data. Will add this discussion.

(R2) "1. Why ...": Asn. 2 is misnamed; we should rather attribute the term "memoryless confounding" to the special
setting of Lemma 1. Asn. 2 is an assumption that it is sufficient to estimate a density ratio that is constant in $s$. For
baseline UCs, marginalized occupancy distributions are understood to be marginalized over an initial state distribution
on the baseline UC.

(R2) "2. While ...": Lemma 1 (to be renamed "memoryless confounding") is just one simple setting where one can
ensure Asn. 2. A practical example may be blood glucose control for diabetic patients, where $s_t$ is blood glucose, $a_t$ is
insulin, and $u_t$ are unobserved eating/exercise events reasonably modeled by a random arrival process (e.g., Poisson).

(R3) "One, ...": In the paper we reference work that discusses how to choose a reasonable range of $\Gamma$; we will instead
flesh out this discussion into the text for completeness. An analyst would have to justify an upper bound on how
informative of selection an unobserved confounder can be; this can be benchmarked relative to the informativeness of
observed covariates by dropping covariates and looking at the distribution of odds ratios for each covariate.

(R3) "Two, ...": Most approaches to sensitivity analysis require making some untestable assumptions. Instead of
assuming the most unrealistic untestable assumption of *no* unobserved confounding, we handle a case where *there*
*is* unobserved confounding but with *structural restrictions*. Asn. 1 is a structural assumption of ergodicity and is
necessary to make sense of infinite-horizon RL, whether with or without confounding. Asn. 2 assumes structure on how
unobserved state variables interact with observed state and actions. Violations of Asn. 2 also violate Asn. 1. If, for
example, nonstationary unobserved confounding (e.g., a single time point) is more plausible for the domain, then our
approach (and other approaches based on stationarity) may be inapplicable. Will mention this and cite the suggested
Namkoong et al. reference regarding single-time-point nonstationary/finite-horizon confounding.

(R3) "Do these put significant constraints on what the evaluation policy can be?": Not if the MDP is ergodic as is often
assumed for infinite-horizon RL (meaning induced chain is ergodic under any deterministic policy). In infinite-horizon
RL, we usually do not deal with MDPs that induce ergodic chains under one policy but not another. We stated our
Asn. 1 in a minimal way since we only really need this for $\pi_e, \pi_b$ but the spirit is that the MDP is ergodic as common
for infinite-horizon RL. Will add this explanation and the stronger version of ergodic MDP.

(R3) "Three": This is a **mischaracterization**: we provide ***both*** globally optimal and heuristic approaches. We will
clarify this in the final text. Prop. 3 provides a disjunctive program formulation that, as we say on line 184, can be
solved directly using branch-and-bound (e.g., Gurobi). In the experiments, following our conclusion in line 184, we
solve Eq. (10) directly in Gurobi (via branch-and-bound with global optimality certificates on bilinear variables). We
further discuss this in appendix line 735. Alg. 1 is provided as a heuristic to tackle large state spaces, and in Fig. 8 of
the appendix we compare the bounds computed by Alg. 1 vs. Gurobi. We will better advertise these results and clarify.

(R3) "empirical results would benefit from verifying both Assumptions": Definitely; we'll comment on this and
explain. Asn. 1 and 2 both hold by construction of the experimental settings. The chains are ergodic for $\pi_e, \pi_b$ and the
confounders satisfy the sufficient condition in Lemma 1.

(R3) "The related works ...": We'll clarify Zhang & Bareinboim and cite Namkoong et al.

(R4) "relies on the discrete nature of $\mathcal{S}$ (which might be okay) ...": We focus on tabular because it is most illustrative
and is very central to RL, but as Remark 2 and Appendix D.1 show all of our results still apply if $w(s) = \theta^T s$ (where $s$
can be embedded arbitrarily). Tabular is the special case where $\mathcal{S} = \{(1, 0, \ldots, 0), \ldots, (0, \ldots, 0, 1)\}$. Indeed going
beyond tabular in RL always requires some function approximation. Rather than further complicate the text, we propose
to more explicitly flesh out the (mostly straightforward) generalization in the appendix.

(R4) "state more clearly the computational complexity": Each step of Alg 1 requires solving two LPs that have size
$|\mathcal{S}|$. LPs are generally considered very easy. We will cite generic theoretical worst-case complexity bounds for LPs,
which while polynomial are not considered representative of their practical difficulty. The branch-and-bound procedure
used by Gurobi is finite-time but not guaranteed to be polynomial. In practice it does very well, solving in seconds for
examples in the paper. We will cite and point to work on the *practical* tractability of integer programming.

[Meta-Review · NeurIPS 2020]

Overall, the reviewers found the paper technically sound, novel, and significant. Personally, I find it quite exciting since it's the first to consider the problem of partial identification in settings with an infinite horizon. My suggestion to improve the paper is to take into account the reviewers' issues and recommendations. After all, my recommendation is "accept."